# A double dissociation between savings and long-term memory in motor learning

**Alkis M. Hadjiosif[1], J. Ryan Morehead[1,2], Maurice A. Smith**[1,3]*

**1** John A. Paulson School of Engineering and Applied Sciences, Harvard University, Cambridge, Massachusetts, United States of America, **2** School of Psychology, University of Leeds, Leeds, United Kingdom, **3** Center for Brain Science, Harvard University, Cambridge, Massachusetts, United States of America

* mas@seas.harvard.edu

**Data Availability Statement:** Individual quantitative observations that underlie the data summarized in the figures and results of this paper can be found in Zenodo (doi: 10.5281/zenodo. 7668262) Extended data and analysis code are maintained on a GitHub repository: https://github. com/AlkisMH/Savings_vs_Long_Term_Memory.

## Abstract

Memories are easier to relearn than learn from scratch. This advantage, known as savings, has been widely assumed to result from the reemergence of stable long-term memories. In fact, the presence of savings has often been used as a marker for whether a memory has been consolidated. However, recent findings have demonstrated that motor learning rates can be systematically controlled, providing a mechanistic alternative to the reemergence of a stable long-term memory. Moreover, recent work has reported conflicting results about whether implicit contributions to savings in motor learning are present, absent, or inverted, suggesting a limited understanding of the underlying mechanisms. To elucidate these mechanisms, we investigate the relationship between savings and long-term memory by experimentally dissecting the underlying memories based on short-term (60-s) temporal persistence. Components of motor memory that are temporally-persistent at 60 s might go on to contribute to stable, consolidated long-term memory, whereas temporally-volatile components that have already decayed away by 60 s cannot. Surprisingly, we find that temporally-volatile implicit learning leads to savings, whereas temporally-persistent learning does not, but that temporally-persistent learning leads to long-term memory at 24 h, whereas temporally-volatile learning does not. This double dissociation between the mechanisms for savings and long-term memory formation challenges widespread assumptions about the connection between savings and memory consolidation. Moreover, we find that temporally-persistent implicit learning not only fails to contribute to savings, but also that it produces an opposite, anti-savings effect, and that the interplay between this temporally-persistent anti-savings and temporally-volatile savings provides an explanation for several seemingly conflicting recent reports about whether implicit contributions to savings are present, absent, or inverted. Finally, the learning curves we observed for the acquisition of temporally-volatile and temporally-persistent implicit memories demonstrate the coexistence of implicit memories with distinct time courses, challenging the assertion that models of context-based learning and estimation should supplant models of adaptive processes with different learning rates. Together, these findings provide new insight into the mechanisms for savings and long-term memory formation.

**Funding:** This work was supported in part by the McKnight Scholar Award to MAS, a Sloan Research Fellowship to MAS, and a grant from NIA (R01 AG041878) to MAS. The funders had no role in study design, data collection and analysis, decision to publish, or preparation of the manuscript.

**Competing interests:** The authors have declared that no competing interests exist.

**Abbreviations:** CUHS, Committee on the Use of Human Subjects; IQR, interquartile range; UDL, use-dependent learning; VMR, visuomotor rotation.

## Introduction

Memories, both declarative and procedural, are easier to relearn than to learn from scratch. This advantage, known as savings, was first appreciated in Hermann Ebbinghaus's seminal work [1], in which he observed that relearning a forgotten list of words was faster than learning a novel list. Savings has since been demonstrated in a plethora of different paradigms, including cognitive tasks in humans [2,3], operant conditioning in animals [4–6], and motor tasks in humans such as saccade adaptation [7], force-field adaptation [8–11], visuomotor adaptation [12–21], and gait adaptation [22,23].

Previous research has generally maintained that savings results from the recall of a previously consolidated long-term memory [14,24,25]. The ability to form enduring long-term memories is one of the most remarkable biological abilities, with some memories lasting a lifetime despite incessant neural plasticity. Elderly Danes recalled mundane details—such as the weather—surrounding the news of their country's invasion and liberation during WWII more than 50 years later [26], and similar examples of memories surrounding shocking and/or consequential events abound [27–29]. People recognize high school classmates decades later [30], tacitly use synthetic grammar rules years after training [31], and identify specific images months [32] or even years [33] later. Long-term memories emerge through consolidation, a process driven by molecular mechanisms that facilitate synaptic activity [34,35] and can be mediated through the hippocampus [36,37]. In motor tasks, the presence or absence of savings itself has often been taken as a litmus test for whether a previously trained memory has been consolidated, at timescales ranging from hours to months [14,24,25]. Consequently, in terms of underlying mechanisms, savings has been further suggested to result from the following: (1) the unmasking of a slower-learning, strong-retention process in a multi-rate learning model [10]; (2) context- or relevance-based switching between such multiple slow processes, each specific to a different memory [38–41]; or (3) reverting to the memory of a previously learned motor plan that was reinforced by success or mere repetition [16,42]. All of these proposed mechanisms focus on savings as the manifestation of a latent, stable, consolidated motor memory that is robust to both interference and the passage of time.

In line with this idea, a recent but influential view has proposed that savings in motor adaptation specifically results from the recall of an explicit strategy [18,43,44], whereas the implicit component of visuomotor learning does not contribute to savings [44]. These studies provided clear evidence for explicit savings, but the paradigms they used elicited little implicit adaptation, limiting the power to assess implicit savings. More recent studies, which elicited greater implicit adaptation, have led to disparate findings, concluding that implicit adaptation is either faster during relearning [19,20] or slower [45]. The Avraham [45] and Albert [20] studies are of particular interest, as they were both designed to isolate implicit adaptation (albeit using different paradigms) and yet reached opposite conclusions.

What could explain this apparent discrepancy? A possibility is that implicit adaptation may not be monolithic. It may consist of distinct components that are relearned at different rates and differentially elicited in these different paradigms. In fact, an intriguing alternative to recall-based mechanisms is that savings arises from changes in learning rate [11,15,23], an idea reinforced by recent work which has demonstrated that the rate at which learning occurs is systematically modulated by specific characteristics of the learning environment. These characteristics include the amount of task-relevant variability present before learning [46,47], the balance between sensory uncertainty and uncertainty about state estimation [48,49], prior exposure to perturbations characterized by a similar covariance structure between learning parameters [50,51], prior exposure to similar motor errors [21,52], and the trial-to-trial consistency of the learning environment [53,54]. In particular, high-consistency environments,

whereby perturbations tend to persist from one trial to the next and thus confer more predictability to the imposed perturbation, can strongly increase learning rates (up to 3×). This is a critical finding as far as the study of savings is concerned: The adaptation paradigms used to study savings usually consist of the same perturbation being active for a large number of trials (usually 60 to 120), resulting in highly consistent errors, which would in turn lead to strongly increased learning rates [52–55]. This mechanism could lead to savings by enabling the faster relearning of a short-term memory, as opposed to the reemergence of a long-term, stable memory. This is consistent with recent results indicating that prior experience with high-consistency errors, as opposed to the repetition of successful actions, leads to savings [21].

Here, we compare the mechanisms that lead to savings and those that lead to the formation of stable, long-term motor memories. We hypothesized that dissecting motor adaptation into specific memory components based on temporal persistence could shed light into these mechanisms. We use short 60-s time delays to dissect overall adaptation into 2 components: temporally-volatile adaptation, which would decay during this time delay, and temporally-persistent adaptation, which would survive the delay [56–59]. Since it decays away in just 60 s, temporally-volatile adaptation would not lead to long-term memory that is associated with stability across timescales that are orders of magnitude greater [9,25,60]. Hence, mechanisms leading to long-term memory would be contained within temporally-persistent adaptation, and thus, consolidation-dependent savings would predict faster learning solely for temporally-persistent adaptation. Conversely, if we found faster relearning solely for temporally-volatile adaptation, that would indicate a savings mechanism that is not driven by long-term memory.

Remarkably, we find that savings is driven not by the reemergence of temporally-persistent motor memories, but instead by faster relearning of temporally-volatile memories. We go on to find that these temporally-volatile memories responsible for savings in our paradigm represent implicit, rather than explicit, adaptation. When we measure the long-term retention of the temporally-persistent and temporally-volatile components, however, we find that it is temporally-persistent adaptation, not temporally-volatile adaptation, that leads to long-term memory. Together, these findings demonstrate a clear double dissociation between savings and long-term memory.

## Results

We designed a set of experiments to elucidate the mechanisms for savings and long-term memory and investigate the relationship between them. We began by investigating whether savings, the faster relearning of a previously learned adaptation, is driven by the reemergence of a previously consolidated temporally-persistent memory, or by a propensity for faster acquisition of a transient, temporally-volatile memory. In particular, we created a paradigm to dissect initial adaptation, the washout of adaptation, and savings in readaptation into temporally-persistent and temporally-volatile components. We first investigated the dynamics by which temporally-persistent and temporally-volatile memories decay during a washout period following initial adaptation, as savings can arise from the incomplete washout of a component of adaptation [10,61]. This allowed us to compare the rates of unlearning for temporally-persistent and temporally-volatile adaptation during washout, and critically, to measure the initial value of both temporally-persistent and temporally-volatile memories prior to readaptation, so that savings could be accurately assessed for both. We next examined how savings depends on temporally-persistent and temporally-volatile memories by measuring savings separately for these 2 components of motor adaptation, allowing us to determine whether one of these memories is specifically responsible for savings. We then investigated whether long-term memory, measured as the retention of a previously trained adaptation 24 h later, is associated with the temporally-persistent or the temporally-volatile component of adaptation.

## Measuring temporally-volatile and temporally-persistent contributions to savings

For Experiments 1 and 2, we recruited $N = 40$ subjects and trained them on a 30˚ visuomotor rotation (VMR) [12,13,15–18,62–64] (**Fig 1A and 1B**). After 80 trials of initial training, subjects were tested for savings after either a short (40-trial) washout period, which was previously reported to be sufficient for the washout of overall VMR adaptation [15], or a longer (800-trial) washout period we employed to effect a more definitive washout. We also used the data from this 800-trial washout to trace out the time course of unlearning for both the temporally-persistent and temporally-volatile components of adaptation; this unlearning would encompass both active unlearning (i.e., relearning the baseline behavior) and natural trial-to-trial decay of adaptation [17,65–68]. Each subject experienced both types of washout duration following training (see **Fig 1C**). In Experiment 1 ($N = 20$), the short washout period was presented first and the long washout period second. In Experiment 2 ($N = 20$), this order was flipped (**Fig 1C**, for a detailed description see Materials and methods).

Throughout these experiments—during both learning and washout—we occasionally inserted 1-min delays which would allow for temporally-volatile adaptation to decay. Since the 1-min delays we imposed amount to 2.5 to 4× the time constant for decay of temporally-volatile adaptation [56–58], approximately 95% decay of volatile adaptation would be expected, effectively isolating temporally-persistent adaptation. We operationally defined temporally-persistent adaptation as the adaptation measured during the post-delay trial and temporally-volatile adaptation as the difference between overall adaptation (itself taken as the average adaptation in the 2 preceding and 2 proceeding non-delay trials) and temporally-persistent adaptation (**Fig 1D**, also see Materials and methods).

## Temporally-persistent adaptation washes out more slowly than overall adaptation

The data from the long, 800-trial washout period allowed us to carefully examine the time course of unlearning for both the overall adaptation and for the temporally-persistent component of it. Analysis of the washout curves revealed that overall adaptation displayed rapid unlearning; however, persistent adaptation (circles in **Fig 2A**) was unlearned much more slowly. We found that by trials 16 to 25, labeled as "early washout" in **Fig 2A**, overall adaptation had already dropped below 10% of the pre-washout asymptotic adaptation level, whereas about 40% of pre-washout persistent learning remained. By trials 51 to 150, labeled as "mid washout," overall adaptation had dropped below 3%, whereas about 20% of persistent adaptation still remained (**Fig 2A**, see inset). Correspondingly, we found the retention of persistent learning to be significantly greater than overall learning in both early washout (t(23) = 4.8, $p = 6.9 \times 10^{-5}$ and mid washout periods (t(39) = 8.5, $p = 1.9 \times 10^{-10}$). To quantify the rate of unlearning during washout for both overall and persistent adaptation, we fit single exponential decay functions to the washout data (see Materials and methods). This revealed the time constants for unlearning to be 6-fold slower for temporally-persistent adaptation than for overall adaptation (median time constant estimated using bootstrap: 106.0 trials, interquartile range (IQR) [92.6 to 121.9] versus 17.4 trials, IQR [15.1 to 20.1], $p < 10^{-4}$, **Fig 2A**), in line with the higher retention we observed in the early- and mid-washout data.

As a minor point, we also noticed that unlearning curves for temporally-persistent adaptation display somewhat greater retention during the early and mid-washout in Experiment 1 compared to Experiment 2 (**Fig 2A**). This might reflect the difference in the amount of training between the 2 conditions, as 2 training blocks (160 trials in total) preceded this washout period in Experiment 1, whereas only a single training block (80 trials) preceded this washout

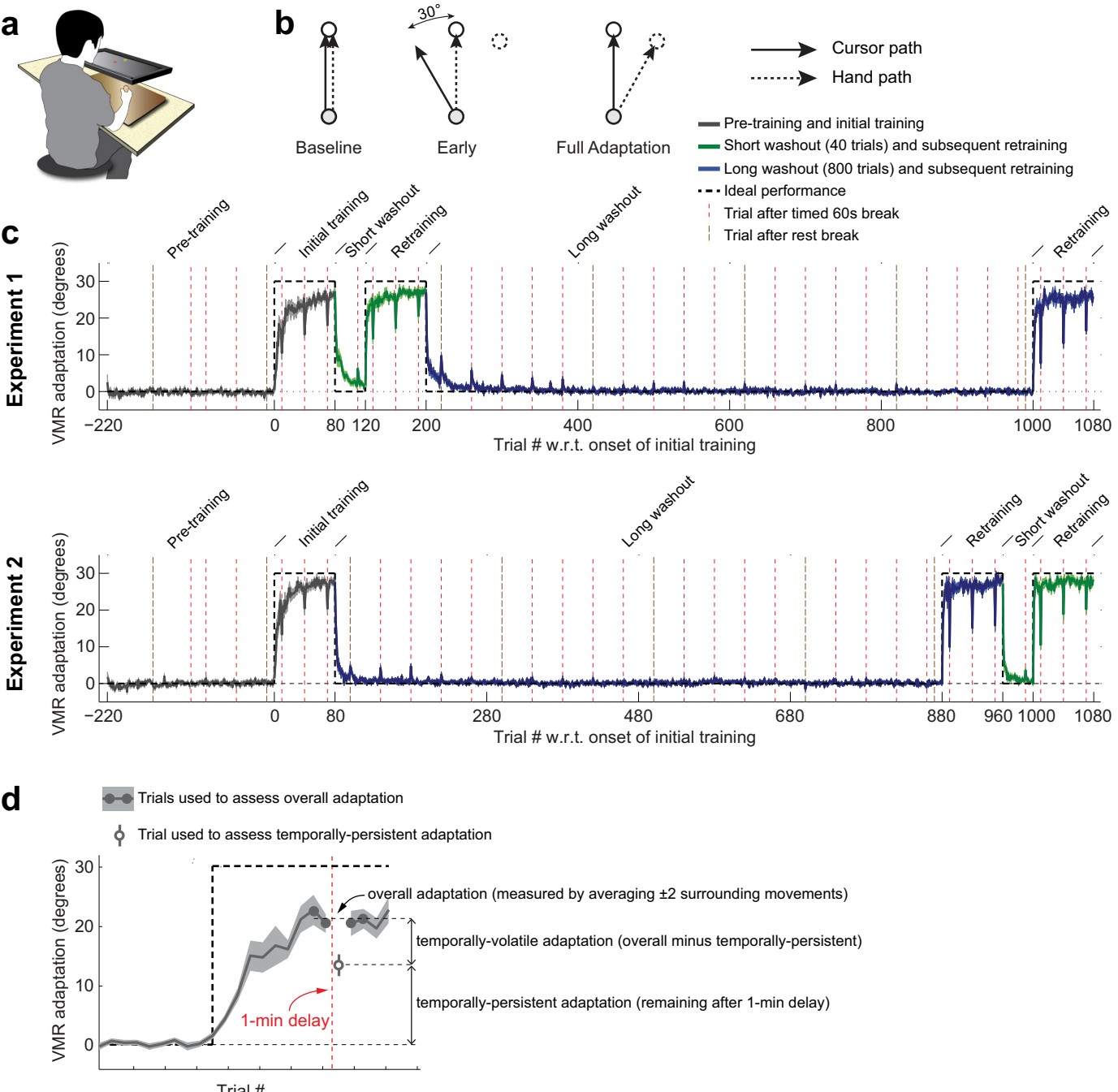

**Fig 1. Experiment setup and training paradigm. (a)** Experiment setup. Subjects made point-to-point reaching movements on a digitizing tablet and received continuous visual feedback on a screen mounted above it. **(b)** VMR training. During baseline (left), the cursor follows the hand motion, whereas during training, cursor motion is skewed by 30° from the hand motion (in this example, counter-clockwise), resulting in a 30° error before adaptation (middle). If full adaptation is achieved, hand motion must completely counter the imposed rotation, corresponding to a 30° clockwise hand motion in this example (right). **(c)** Top: experiment schedule and raw data for Experiment 1. There were 3 phases: a baseline period followed by the initial 80-trial VMR training (average adaptation level shown in gray); a short, 40-trial washout period followed by retraining (green); and a long, 800-trial washout period followed by another 80-trial retraining session (blue). Red dashed vertical lines indicate trials conducted after 60-s delays to isolate temporally-persistent adaptation. Brown dashed vertical lines indicate trials following rest breaks. Note that, during the washout periods, adaptation peaks on these delay and rest break trials, illustrating a slower washout for temporally-persistent vs. temporally-volatile adaptation. Bottom: same but for Experiment 2, where the long, 800-trial washout period came first. Error bars indicate SEM. Underlying data supporting this panel can be found in file Exp_1_2_data.mat. **(d)** Dissection of adaptation into temporally-persistent and temporally-volatile components. Throughout the experiment, we used 60-s delays to allow temporally-volatile adaptation to decay. The amount of adaptation on the trial following such a delay (open gray circle) was taken as a measure of temporally-persistent adaptation, whereas the average amount of adaptation 2 trials before and 2 trials after this post-delay trial (filled gray

circles) was taken as a measure of overall adaptation. Temporally-volatile adaptation was operationally defined as the difference between overall and persistent adaptation.

in Experiment 2, due to the condition balancing (see **Fig 1**). In summary, we found that temporally-persistent adaptation is unlearned at a considerably slower rate than overall adaptation.

### Residual adaptation prior to the onset of retraining

One consequence of the slower unlearning of persistent compared to overall adaptation is that, while the washout of overall adaptation can appear complete after a short 40 to 100 trial per direction washout period [15,16,18,24], substantial temporally-persistent adaptation can nevertheless remain. This suggests that longer washout periods may be required to examine savings independent of the effect of residual temporally-persistent adaptation and that measuring this residual adaptation prior to relearning may facilitate a better understanding of relearning behavior.

When we measured the residual overall and persistent adaptation before the onset of retraining, we found significant levels of both overall and temporally-persistent adaptation for the 40-trial washout but no significant residuals of the previous adaptation following the 800-trial washout period. In particular, we found small but significant residuals for overall adaptation at the end of the 40-trial washout periods, around 5% of pre-washout levels (1.34 ± 0.37°, t(39) = 3.6, $p$ = 0.00087 for Experiments 1 and 2 combined, with positive values indicating adaptation in the direction of the previously imposed VMR, **Fig 2B**). Note that

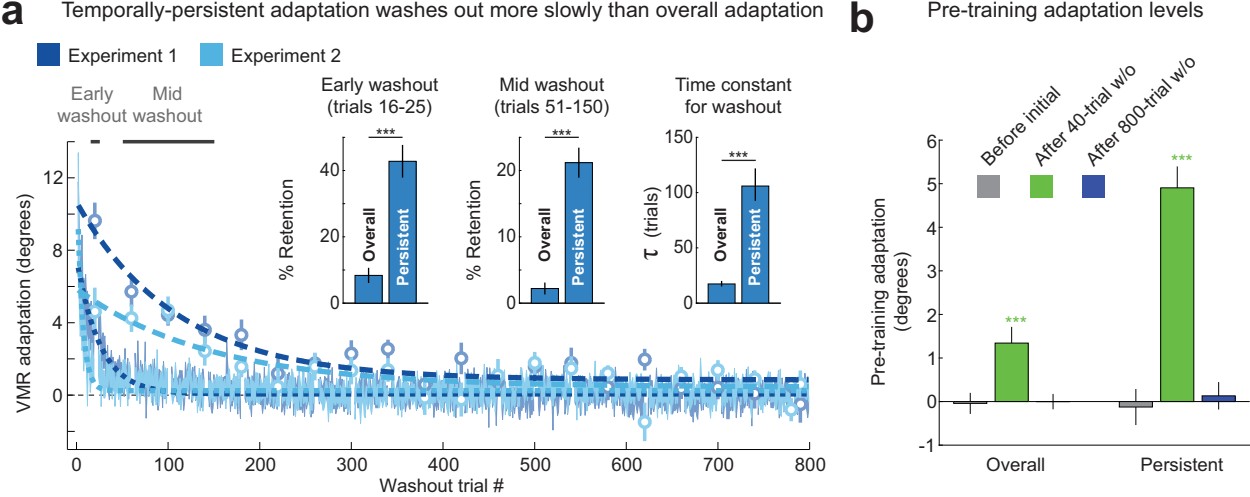

**Fig 2. Temporally-persistent adaptation washes out more slowly than overall adaptation. (a)** Washout curves for the overall adaptation (shading indicates mean ± SEM) and temporally-persistent adaptation (circles) for both Experiment 1 (blue) and Experiment 2 (light blue), illustrating the contrast between rapid washout for overall adaptation and slower washout for temporally-persistent adaptation. The thick dashed or dotted lines indicate exponential fits. Inset: Three different measures of washout for overall vs. persistent adaptation. Retention expressed as a percentage of asymptote adaptation after 16–25 trials (left) or 51–150 trials (center) indicates slower washout for temporally-persistent compared to overall adaptation. Time constants for the washout curves (right) also show slower washout for temporally-persistent adaptation. *$p < 0.05$, **$p < 0.01$, ***$p < 0.001$. **(b)** Residual adaptation before initial training (gray, left bar in each cluster), at the end of the 40-trial washout period (green, middle bar) and at the end of the 800-trial washout period (blue, right bar). The data show that the 40-trial washout period leaves a significant amount of temporally-persistent adaptation and a smaller but also significant amount of overall adaptation. Consequently, retraining after only 40 washout trials starts from a nonzero baseline. *$p < 0.05$, **$p < 0.01$, ***$p < 0.001$. Underlying data supporting this figure can be found in file Exp_1_2_data.mat.

these residuals were related to previous adaptation rather than a movement direction bias because both Experiments 1 and 2 were balanced, with 10 participants trained with clockwise, and 10 with counter-clockwise VMRs for each experiment. The residuals were even larger for temporally-persistent adaptation, in line with the substantially slower unlearning of temporally-persistent adaptation compared to overall adaptation we observed. In particular, we found that the residual persistent adaptation before the end of the 40-trial washout was around 25% of pre-washout persistent adaptation ($4.91 \pm 0.49°$ for Experiments 1 and 2 combined, t(39) = 10.0, $p = 2.7 \times 10^{-12}$). In contrast, the 800-trial washout period was sufficient to bring both overall adaptation and temporally-persistent adaptation back to baseline, with measured residuals of only 0% to 2% of pre-washout levels on average. These residuals were not consistently in the direction of the pre-washout adaptation and were not statistically significant. Overall adaptation at the end of the 800-trial washout was $-0.00 \pm 0.18°$ (t(39) = −0.01, $p = 0.9955$), whereas temporally-persistent adaptation was $0.13 \pm 0.32°$ (t(39) = 0.44, $p = 0.6772$), as shown in **Fig 2B**.

These results show that a prolonged washout period is required to eliminate residual temporally-persistent adaptation. As washout periods in previous experimental work on savings [15,16,18,24,64] are typically much shorter than the 800-trial washout period we examined, it is likely that the savings observed in these studies is, at least in part, driven by interactions between different components of adaptation that were not fully washed out prior to retraining —apparent savings—as suggested in Smith and colleagues [10]. In order to examine faster relearning that is not contaminated by such interactions—that is, examine true savings—one should ideally eliminate residual levels of overall, temporally-persistent, and temporally-volatile adaptation, or, at least, take these residual levels into account.

## Savings is present even when previous adaptation is completely washed out

To investigate savings for overall and temporally-persistent adaptation, we compared the learning curves for retraining and initial training, shown in **Fig 3A and 3B** (gray: initial training; green: retraining after 40 washout trials; blue: retraining after 800 washout trials). We found that the adaptation levels achieved in the early adaptation period (trials 8 to 12 after perturbation onset, excluding trial 10 which was after a time delay), when learning was most rapid, were noticeably higher for relearning ($24.6 \pm 0.7°$ overall; $24.9 \pm 0.6°$ and $24.4 \pm 1.0°$ after the 40-trial and 800-trial washout periods, green and blue lines, respectively, with data combined across Experiments 1 and 2 in all cases) compared to initial training ($18.8 \pm 1.2°$, gray; $p < 10^{-5}$ compared to relearning after either washout period or both periods combined). Because pretraining adaptation levels were not identical across conditions as shown in **Fig 2B** ($1.34 \pm 0.37°$, $-0.00 \pm 0.18°$, and $-0.04 \pm 0.24°$, after short washout, long washout, and before initial adaptation), we normalized data to quantitatively compare learning and relearning curves independent of the effect of this residual pretraining adaptation. Specifically, we subtracted the pretraining adaptation separately for overall and temporally-persistent adaptation, and normalized each baseline-subtracted learning curve by the distance between baseline and the ideal adaptation level (Eq 2, see Materials and methods). These normalized data, plotted in **Fig 3C and 3D** express adaptation levels as a percentage of that required for full adaptation. In particular, normalized early adaptation (trial 10 after perturbation onset) was faster compared to initial adaptation (initial adaptation: $62.8 \pm 4.1\%$ versus relearning: $81.7 \pm 2.5\%$; 40-trial and 800-trial washout data separately: $82.1 \pm 2.1\%$ and $81.4 \pm 3.2\%$, respectively). We defined savings simply as the difference between these normalized adaptation data for the retraining versus the initial learning conditions. The top panel in **Fig 3E and 3F** shows an estimate of this savings measure in the overall adaptation. We find statistically significant savings for early

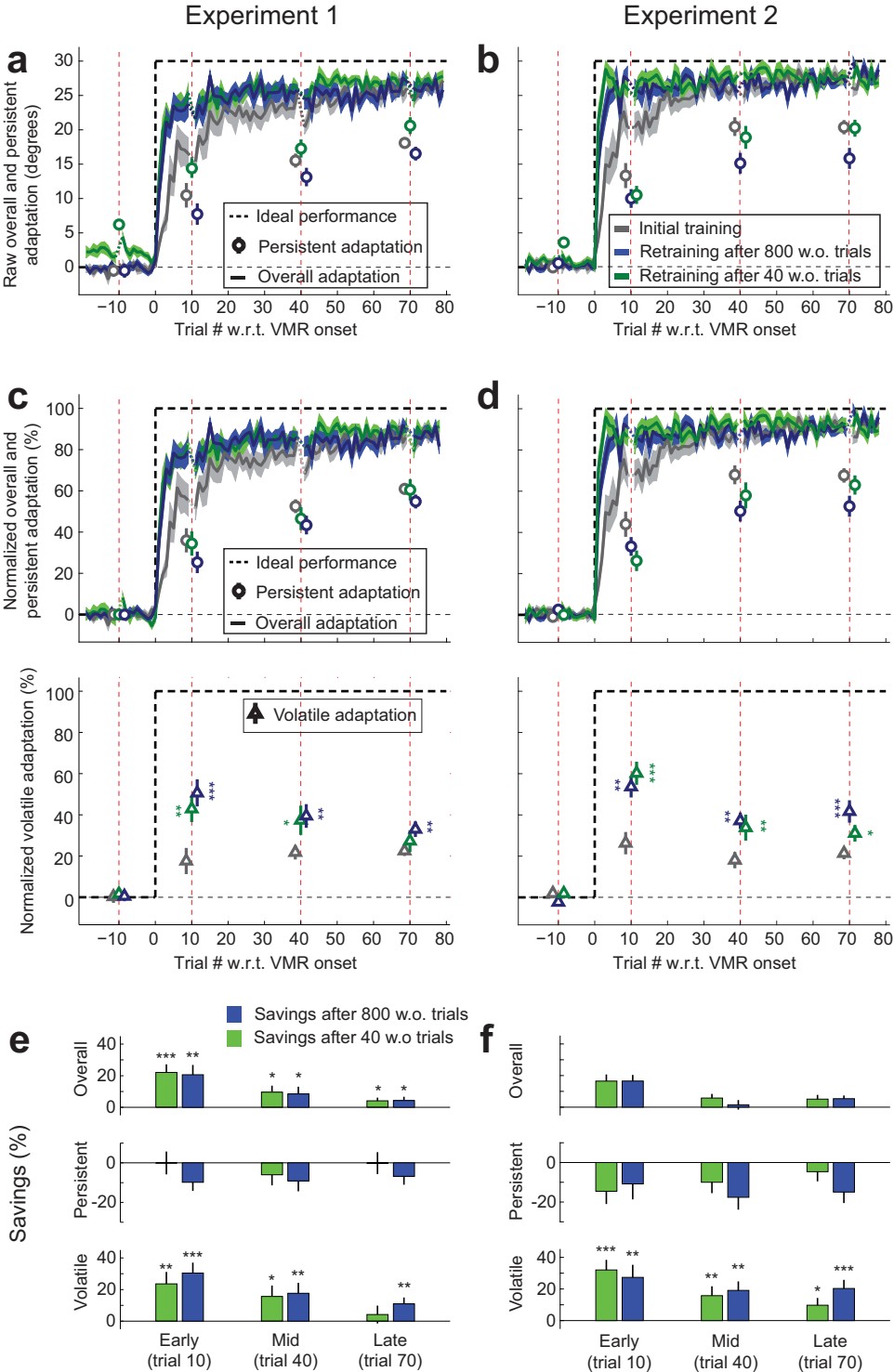

**Fig 3. Temporally-volatile adaptation displays savings, but temporally-persistent adaptation does not. (a, b)**
Learning curves for Experiments 1 and 2, showing overall and persistent adaptation (solid lines and circles,
respectively). Persistent adaptation during training or retraining was assessed after a 60-s delay as shown in Fig 1. The
dashed black line indicates ideal performance and error bars indicate SEM. Overall adaptation (solid lines) is faster for
retraining after both a 40-trial washout period (green) and an 800-trial washout period (blue) when compared to initial
adaptation (gray). In contrast, persistent adaptation (circles) is not faster during retraining than initial training. Note
that performance during the retraining period following a 40-trial washout (green) can be inflated because this

washout is often incomplete for both overall and persistent learning, as indicated by nonzero pretraining levels as also shown in **2b**. Washout for both overall and persistent adaptation is complete for the 800-trial data (blue) in both Experiments 1 and 2 resulting pretraining levels that are essentially zero in all cases. Vertical red dashed lines indicate trials following 1-min delays. **(c, d)** Normalized overall, persistent, and volatile adaptation components in Experiments 1 and Experiment 2. Upper panels show the normalized learning curves for overall and temporally-persistent adaptation. Here, the raw data shown in A were normalized by subtracting out pretraining adaptation levels so that training-related changes in performance can be directly assessed and scaling ideal performance to 100% (see Materials and methods, Eq 2). Whereas the overall adaptation is faster for retraining after both washout periods, persistent adaptation is not. Lower panels show the normalized learning curves for temporally-volatile adaptation. Temporally-volatile readaptation is consistently higher than initial adaptation, especially for the early (trial 10) time point, where savings should be most pronounced and where readaptation is between 2–3 times higher. **(e, f)** Savings for overall, temporally-persistent, and temporally-volatile adaptation in Experiments 1 and 2, defined as the difference between the normalized adaptation metrics displayed in panels **c and d** for relearning vs. initial learning (see Materials and methods, Eq 3). Positive values for savings indicate faster relearning. We find substantial and significant savings especially early during training (trial 10) for both overall and temporally-volatile adaptation, but not for temporally-persistent adaptation, suggesting that savings is driven by temporally-volatile adaptation. Underlying data supporting this figure can be found in file Exp_1_2_data.mat.

adaptation (trial 10; savings of 18.9 ± 3.3% of the ideal adaptation, t(39) = 5.8, $p = 5.4 \times 10^{-7}$ [40-trial washout data: 19.3 ± 3.3%, t(39) = 5.8, $p = 4.4 \times 10^{-7}$; 800-trial washout data: 18.6 ± 3.6%, t(39) = 5.1, $p = 4.1 \times 10^{-6}$). Inspection of the overall adaptation data in the top panels of **Fig 3C and 3D** reveals that both the readaptation and initial adaptation curves asymptote near the ideal adaptation level, meaning that the room for improvement, and thus the capacity for savings, is reduced as training proceeds. In line with this observation, the savings we observed for mid (trial 40) and late (trial 70) overall adaptation were smaller than the savings observed for early (trial 10) adaptation (<10% of the ideal adaptation in all cases). The overall savings we observed at trials 40 and 70 were, however, statistically significant (t(39) = 2.6, $p = 0.0063$ for trial 40 and t(39) = 3.3, $p = 0.0011$ for trial 70), as shown in **Fig 3E and 3F**.

## Savings arises from the faster acquisition of temporally-volatile memories

Remarkably, we found that the savings observed in overall adaptation was due to temporally-volatile learning. We calculated savings for temporally-volatile adaptation (bottom row in **Fig 3E and 3F**) based on normalized volatile adaptation (bottom row of **Fig 3C and 3D**), which was computed as the difference between the normalized overall and normalized persistent adaptation (top row of **Fig 3C and 3D**). We found that volatile adaptation during early training (trial 10) was 2- to 3-fold faster for retraining than for initial training after both the short and long washout periods in both experiments (as shown in the bottom row of **Fig 3C and 3D**). Specifically, we found that trial 10 volatile readaptation was 52.2 ± 3.8% of the ideal adaptation to the 30˚ VMR with data after both types of washout combined versus 22.0 ± 4.2% for initial adaptation, t(38) = 6.2, $p = 1.4 \times 10^{-7}$ (readaptation for 40-trial washout data: 51.3 ± 4.5%, t(37) = 5.7, $p = 9.0 \times 10^{-7}$ for savings; readaptation for 800-trial washout data: 52.1 ± 4.1%, t(38) = 5.6, $p = 9.9 \times 10^{-7}$ for savings). This indicates substantial, statistically significant savings in temporally-volatile adaptation as illustrated in the bottom row of **Fig 3E and 3F**.

## Savings does not arise from the rapid reemergence of temporally-persistent memories

Intriguingly, the clear pattern of savings we found in the learning curves for overall and temporally-volatile adaptation was absent for temporally-persistent adaptation. In only 1 of the 4 conditions in Experiments 1 and 2 (readaptation after a 40-trial washout in Experiment 1) was the unnormalized temporally-persistent adaptation even nominally higher during relearning

than initial adaptation, and in that condition the readaptation built upon a substantially higher pretraining level than the corresponding initial training condition (**Fig 3A**). When pretraining levels of persistent adaptation were taken into account by normalizing the learning curves, we found that relearning for temporally-persistent adaptation was nominally slower, rather than faster, than initial learning in all 4 conditions as shown in **Fig 3C and 3D**. Specifically, early (trial 10) savings were, on average −10.0 ± 4.3% of the ideal persistent adaptation, t(38) = −2.3, $p$ = 0.99 for savings (40-trial washout data: −7.3 ± 4.4%, t(37) = −1.7, $p$ = 0.95; 800-trial washout data: −10.3 ± 4.5%, t(38) = −2.3, $p$ = 0.99) as shown in **Fig 3E and 3F**. The temporally-persistent adaptation measured 40 and 70 trials into the training period for the combined 40-trial and 800-trial washout data displays results similar to trial 10 adaptation, with a tendency towards anti-savings (slower readaptation) (t(39) = −3.4, $p$ = 1.00 for trial 40 and t(38) = −2.2, $p$ = 0.98 for trial 70). The absence of savings in temporally-persistent adaptation stands in stark contrast to the high levels of savings observed in temporally-volatile adaptation, suggesting that overall savings arises from the former, but not the latter. Thus, our result indicates that savings arises from the faster relearning of volatile memories, rather than the re-manifestation of persistent memories.

## Temporally-persistent memories display anti-savings

Based on recent work that reported anti-savings for implicit motor adaptation [45], we asked, in a post-hoc analysis, whether temporally-persistent adaptation consistently displayed the slowed relearning that would constitute anti-savings. This analysis revealed that, relearning for persistent adaptation was, in fact, significantly slower than initial learning (t(38) = −2.3, $p$ = 0.0247, 2-tailed paired $t$ test), based on the trial 10 the data from both washout periods combined. This anti-savings was most clear in the long 800-trial washout data, which allowed us to examine savings without any effects of residual temporally-persistent adaptation (t(38) = −2.3, $p$ = 0.0276, 2-tailed paired $t$ test). Savings at trial 10 after the short incomplete washout was also nominally negative but, in this case, not significantly so (t(37) = −1.7, $p$ = 0.1073, 2-tailed paired $t$ test). Analysis of temporally-persistent adaptation at trials 40 and 70 provides consistent results, with statistically significant anti-savings observed for the combined data from the 40 and 800-trial washout periods (t(39) = −3.4, $p$ = 0.0017 at trial 40; t(38) = −2.2, $p$ = 0.0359 at trial 70, 2-tailed paired $t$ tests) and also for the 800-trial washout data analyzed in isolation (t(39) = −3.3, $p$ = 0.0022 at trial 40; t(38) = −3.1, $p$ = 0.0035 at trial 70, 2-tailed paired $t$ tests). Accordingly, the 40-trial washout data analyzed in isolation showed mixed results at these individual time points (t(39) = −2.1, $p$ = 0.0417 at trial 40; t(38) = −0.7, $p$ = 0.5053 at trial 70, 2-tailed paired $t$ tests). However, when the data combined across all time points are used, we find individually significant anti-savings for both washout periods (t(39) = −5.0, $p$ = 0.000012, for the 800-trial data; t(39) = −2.6, $p$ = 0.0131 for the 40-trial data). In sum, the negative savings results we observe in the 800-trial and 40-trial washout data are similar, but it appears that the 800-trial result is somewhat clearer, possibly because the 40-trial washout data suffer from incomplete washout of the initial adaptation before relearning. Overall, our data show a conspicuous absence of savings in the relearning of temporally-persistent adaptation in all conditions we examined, instead showing anti-savings despite robust savings in the relearning of temporally-volatile adaptation.

   Although a small effect, we found it interesting that anti-savings was somewhat more consistently observed following the 800-trial washout condition than the 40-trial condition, suggesting that the prolonged repeated execution of the same no-rotation trials that constitute the washout period might make anti-savings more consistent. Indeed, the strengthening of an action following repeated execution, often termed use-dependent learning (UDL) [69,70], can

manifest in reaching in the form of a directional bias toward its direction [70–72]. Interestingly, the expected bias toward the baseline no-rotation movement direction following washout would oppose the movement direction changes associated with VMR relearning, and thus act in the direction of anti-savings to reduce relearning.

However, it is critical to note that (1) because we are examining savings, the slowed relearning that constitutes anti-savings refers to slower than initial learning; and (2) that the initial learning period in our experiments was also preceded by a prolonged period of the execution of repeated no-rotation trials, which would likewise elicit a UDL effect. The key question would not, therefore, be whether a UDL effect might slow relearning following the 800-trial washout period, but whether such an effect would show a meaningful size increase between the 220-trial duration of the no-rotation baseline period that precedes initial learning and the 800-trial no-rotation washout period that precedes the 800-trial relearning condition? However, the available literature on how UDL effects increase with the number of repeated trials suggests that this is unlikely. Studies examining UDL effects in reaching movements showed effects after only 1 to 15 trials ([71], Exp 3; [70]), and the one study that looked at the time course for UDL effects beyond 15 trials, found effects that asymptoted between 50 and 150 trials ([71], Fig 4), which is smaller than the duration of the 220-trial baseline that preceded initial learning in our experiment. This suggests that UDL effects, if they indeed affect VMR training in our study, would do so equally for both initial learning, which was preceded by 220 no-rotation trials, and relearning following long-washout, which was preceded by 800 no-rotation trials. Consequently, UDL effects should have little effect on the difference between these learning curves and thus on the savings we measure after 800 washout trials, and are, therefore, unlikely to explain the temporally-persistent anti-savings we observed.

However, the one paper that looked at the effect of repeat duration [71] did not study UDL training periods as long as 800 trials (540 trials was their maximum) and investigated UDL outside the context of VMR adaptation. We thus performed an additional experiment (Experiment S1) to determine whether the differences between the number of no-rotation reaches before initial learning (220 trials) versus before the long-washout relearning (800 trials) might explain the anti-savings we observe. Experiment S1 examined initial learning after a baseline of 800 rather than 220 trials to match the duration of the action selection history of the 800-trial washout before relearning.

If the reduction in temporally-persistent relearning with respect to initial learning we observed were indeed due to the longer 800 trial movement/action selection history, we would expect the initial temporally-persistent learning in this new dataset to match the slowed temporally-persistent relearning from the 800-trial washout condition rather than the initial temporally-persistent learning that we previously observed after 220 baseline trials. However, the results from Experiment S1 instead show that the initial temporally-persistent learning following 800 baseline trials in this new dataset was a closer match to the initial temporally-persistent learning that was previously observed after 220 baseline trials, as this dataset did not show the slower initial temporally-persistent learning that would be predicted by increased UDL following 800 rather than 220 trials (trial 10 learning: $47.1 \pm 9.6\%$ for initial learning in the new data versus $40.1\% \pm 4.2\%$ for initial learning in the previous data, $t(49) = -0.8$, $p = 0.45$, see **S1 Fig**). Instead, as shown in **S1 Fig**, the data are in line with the Verstynen and Sabes data [71] whereby UDL effects asymptote before 220 trials. In line with this prediction, the temporally-persistent relearning following 800-trial washout trials observed in the previous data was also significantly slowed compared to this new 800-trial baseline data with significant anti-savings observed when the data from trials 10, 40, and 70 were averaged together ($t(50) = -3.2$, $p = 0.0014$) and also when the data from these time points were analyzed separately ($t(50) = -2.2$, $p = 0.017$ at trial 10; $t(50) = -2.5$, $p = 0.0086$ at trial 40; $t(50) = -2.2$, $p = 0.0162$ at trial

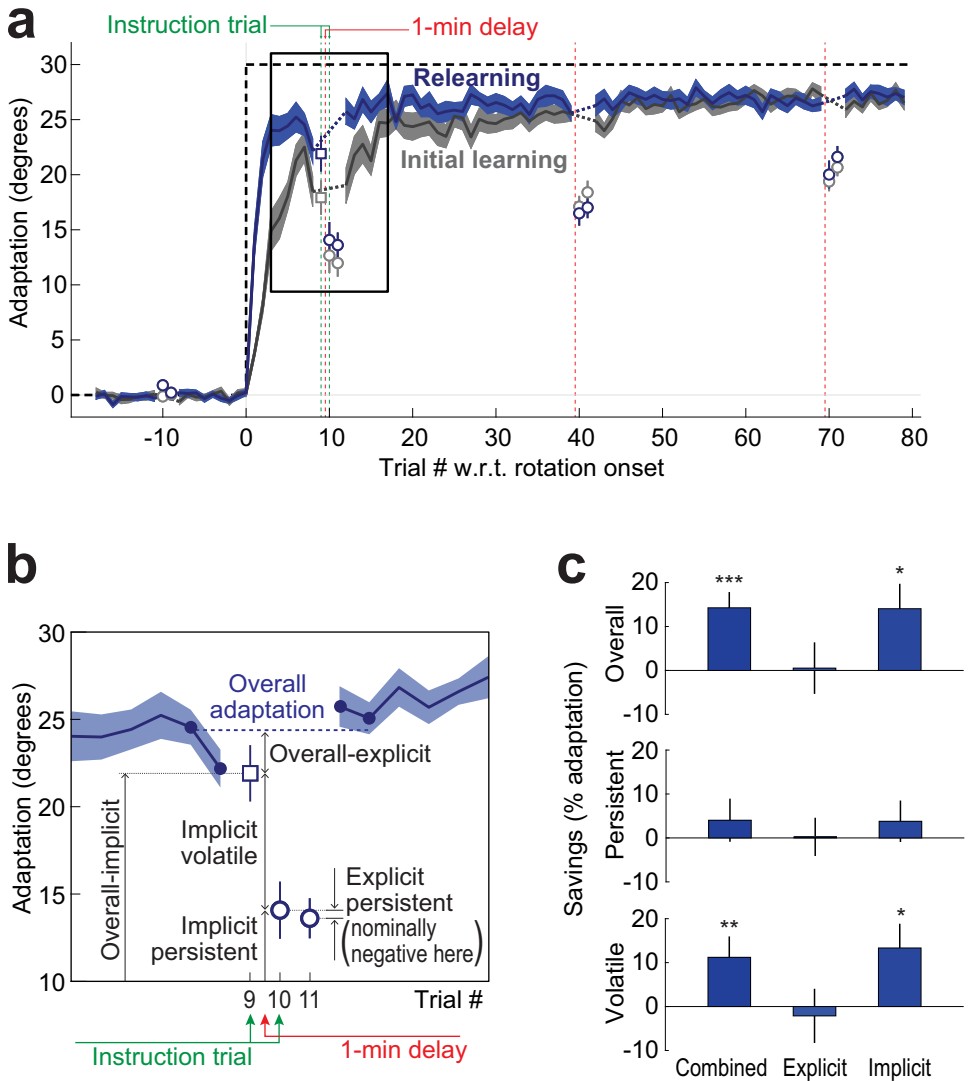

**Fig 4. Temporally-volatile savings are due to an implicit adaptation component. (a)** Learning curves for Experiment 3, showing overall adaptation (solid lines), overall implicit adaptation (square) and persistent adaptation (circles). Gray denotes initial learning, blue denotes relearning. Shading indicates SEM. **(b)** Diagram illustrating measurements of specific components of implicit and explicit learning for Experiment 3, based on a zoomed-in view of the blue retraining data from panel **a**. Instructions 1 trial before and 1 trial after the 1-min delay (trials 9 and 10) allow the direct measurement of overall-implicit and implicit-persistent components, respectively. This allows measurement of implicit-volatile as the difference between them, and overall-explicit adaptation as the difference between overall adaptation and overall-implicit, where overall adaptation is measured as the average amount of adaptation 2 trials before the first instruction trial and 2 trials after the last persistent-only trial (i.e., trials 7, 8, 12, and 13; blue filled circles). Both instruction trials have no visual feedback to avoid per-trial learning leading to posttrial recovery of adaptation. The following trial (trial 11) thus allows direct measurement of combined implicit and explicit persistent adaptation, therefore, the difference between trial 11 and trial 10 measures explicit-persistent. Trial 11 also reintroduces visual feedback, allowing adaptation to recover by the next trial. **(c)** Savings in overall, temporally-persistent, and temporally-volatile adaptation in Experiment 3, both for combined implicit and explicit adaptation (left column) and broken into implicit and explicit components. In line with Experiments 1 and 2, data show overall and volatile, but not persistent, savings for combined (implicit + explicit) adaptation. Further dissociation into implicit and explicit components reveals this savings is due to implicit, temporally-volatile adaptation. Error bars indicate SEM. *$p < 0.05$; **$p < 0.01$. Underlying data supporting this figure can be found in file Exp_3_data.mat.

70). These findings suggest the UDL effects cannot explain the 800-trial temporally-persistent anti-savings we observe.

As a side note, if UDL effects were, on the other hand, somewhat lower after 40 trials (the short washout period duration) compared to 220 trials (the baseline period duration) [71], it would lead to a reduction in the amount of UDL-induced slowing for relearning after 40 washout trials compared to the initial learning. This would suggest that, if anything, we underestimated temporally-persistent anti-savings in the 40-trial washout data rather than overestimating it in the 800-trial washout data, perhaps contributing to the less consistent results when this condition was considered in isolation.

## Temporally-volatile savings arise from implicit adaptation

Previous research associated savings in visuomotor adaptation with the rapid recall of explicit strategies, rather than faster implicit adaptation [18,43,44]. This led us to investigate the contributions of implicit and explicit processes in the temporally-volatile savings we observed in our paradigm. We thus ran Experiment 3 ($N$ = 40), which consisted of two 80-trial learning episodes separated by 800 washout trials. We dissected savings into implicit and explicit components using special instruction trials that prompted participants to disengage any explicit strategy by aiming their hand directly to the target [44,73–78]. These instructions were presented immediately before and after the first (trial 10) 60-s time delay following the onset of the VMR in both initial learning and relearning and allowed us to dissect adaptation into 4 subcomponents: implicit-persistent, implicit-volatile, explicit-persistent, and explicit-volatile (**Fig 4B**, see Materials and methods for details).

In line with our findings in Experiments 1 and 2, we found savings for overall and volatile adaptation (14.3 ± 3.6%, t(39) = 4.0, $p$ = 0.00014 and 11.2 ± 4.8%, t(37) = 2.4, $p$ = 0.0119, correspondingly) but not persistent adaptation (4.0 ± 4.9%, t(38) = 0.8, $p$ = 0.21). Dissection of savings into explicit and implicit components revealed savings for both overall implicit and implicit-volatile adaptation (14.1 ± 5.7%, t(38) = 2.5, $p$ = 0.0088 and 13.3 ± 5.5%, t(37) = 2.4, $p$ = 0.0104, correspondingly) but not explicit-volatile adaptation (−2.1 ± 6.2%, t(37) = −0.3, $p$ = 0.63) or any of the persistent subcomponents (implicit-persistent: 3.8 ± 4.7%, t(38) = 0.8, $p$ = 0.21; explicit-persistent: 0.2 ± 4.4%, t(38) = 0.1, $p$ = 0.48). This finding suggests that overall savings were driven by the implicit and temporally-volatile component of adaptation, in turn suggesting that the temporally-volatile savings we observed in Experiments 1 and 2 predominantly reflect an implicit process rather than an explicit strategy. That the volatile component observed in Experiments 1 and 2 is primarily implicit is not surprising: First, it is unclear why an explicit strategy could be temporally-volatile to the point of being largely or completely forgotten after a short 1-min delay. In fact, our recent work indicates that explicit adaptation displays essentially no temporal volatility, with over 95% stability across 1-min delays [79]. Second, our paradigm elicited scant explicit adaptation (likely due to elements of our experiment design aimed at inducing implicit learning such as the use of point-to-point (rather than shooting) movements, the lack of aiming instructions, the lack of markers that could aid off-target aiming, and the presence of low-latency online feedback [76,80–83]) and without substantial explicit adaptation we lacked power for measuring explicit savings.

## Dissecting long-term memory in visuomotor adaptation

We next investigated whether the ability to dissect motor learning into temporally-persistent and temporally-volatile components could shed light on the mechanisms for the formation of long-term memories. To accomplish this, we examined the relationship between the levels of temporally-persistent and temporally-volatile learning observed after initial training and the

amount of retention observed 24 h later (Experiment 4). After a baseline period, we trained 25 participants on a 30° VMR for 120 trials. After this initial training, they were tested for temporally-persistent adaptation as present after a rest break (average break duration: 125 ± 8 s, which would let >99% of temporally-volatile adaptation decay based on a time constant of approximately 20 s). The above measurements were then repeated, with participants retrained for 60 trials, and retested for temporally-persistent adaptation (the average of these 2 measurement sessions was used to quantify temporally-persistent adaptation for each individual). Participants then returned the following day to be tested for retention (**Fig 5A**, see Materials and methods).

We found that, the overall adaptation measured late in training (the last 20 trials) in Experiment 4 was similar to that observed in Experiments 1 and 2 (27.4 ± 0.3° for Experiment 4 versus 26.4 ± 0.6° and 27.7 ± 0.5° for Experiments 1 and 2, see **Fig 5B**). Similarly, the persistent component of adaptation was also similar across the 3 experiments (16.7 ± 1.0° for Experiment 4 versus 18.1 ± 0.7° and 20.4 ± 1.0° for Experiments 1 and 2, see **Fig 5B**), suggesting that the somewhat longer training duration in Experiment 4 had little effect on either overall or temporally-persistent adaptation. When examining long-term memory, retained 24 h after training, we found that participants retained 8.9 ± 1.1° of the trained 30° rotation (orange bar in **Fig 5B**). This corresponded to 32.4 ± 4.2% of the overall learning and 52.7 ± 5.2% of the temporally-persistent learning from day 1.

## Dissociable effects of temporally-volatile and temporally-persistent adaptation on the formation of long-term memory

To examine whether the dissection of day 1 learning into temporally-persistent and temporally-volatile components could shed light on the mechanism for long-term motor memory formation, we compared the levels of temporally-volatile, temporally-persistent, and overall learning to the amount of 24-h retention for each individual participant. Looking for positive contributions of each component to 24-h retention (using linear regression with regression coefficients restricted to be positive), we found no significant relationship between overall learning on day 1 and 24-h retention on day 2 ($r = +0.14$, $F(23,1) = 0.4$, $p = 0.51$). However, we found a highly significant positive relationship between persistent learning on day 1 and 24-h retention (slope = 0.80, $r = +0.71$, $F(23,1) = 22.9$, $p = 0.00008$). In contrast, we found no positive relationship between volatile learning on day 1 and 24-h retention; in fact, the best fit slope was zero ($r = 0.0$, $F(23,1) = 0$, $p = 1$), as the best fit slope without restricting regression coefficients to positive values would have been negative. This indicates that temporally-volatile learning does not lead to 24-h retention, consistent with the fact that volatile learning, by definition, will decay over the course of 1 min. We thus find that, whereas neither overall adaptation nor the temporally-volatile component can predict it, the temporally-persistent component of adaptation, measured only 1 min after training, is able to accurately predict retention 24 h after training.

We next performed a stepwise bivariate regression analysis of how 24-h retention was associated with temporally-volatile and temporally-persistent learning from day 1, as illustrated in **Fig 6A and 6B**. This analysis was particularly important here because temporally-volatile and temporally-persistent learning were not independent across individuals but instead displayed a strong negative relationship such that participants with higher day 1 temporally-volatile learning displayed smaller day 1 temporally-persistent learning and vice versa. This bivariate regression revealed that adding temporally-volatile learning as a second regressor after temporally-persistent learning resulted in no significant improvement in the ability to explain 24-h retention ($R^2$ increased from 49.8% to 51.7% corresponding to a partial $R^2$ of only 3.8%, F

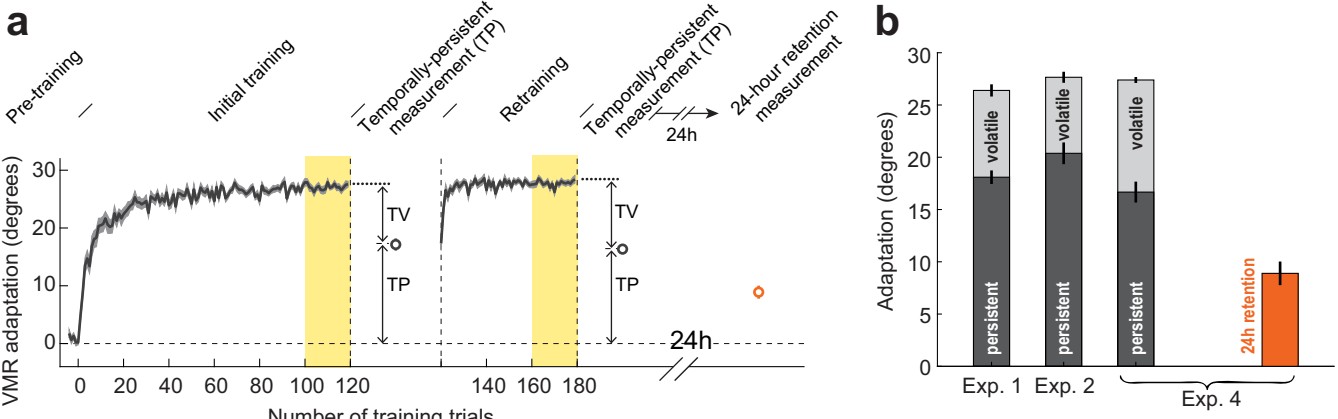

**Fig 5. Measuring temporally-volatile and temporally-persistent components of adaptation and subsequent long-term retention.** (a) Experiment schedule and raw data for Experiment 4. After a baseline period, subjects were trained with a 30° VMR for 120 trials, and were then tested, after a break, for temporally-persistent adaptation, retrained for 60 trials, and then retested after another break. Subjects returned the following day when they were tested for 24-h retention (orange circle). Note that 24-h retention is lower than overall or temporally-persistent adaptation but higher than zero. TP: temporally-persistent; TV: temporally-volatile. Yellow background indicates trials used to measure overall adaptation. Error bars and shading indicate SEM. (b) Comparison of overall, temporally-persistent, and temporally-volatile adaptation from Experiments 1, 2, and 4 with the 24-h retention from Experiment 4. Experiments 1, 2, and 4 display similar levels of persistent and volatile adaptation. Underlying data supporting this figure can be found in file Exp_4_data.mat.

(22,1) = 0.9, $p$ = 0.36). In contrast, adding temporally-persistent learning as a second regressor after temporally-volatile learning resulted in a large improvement in the ability to explain 24-h retention ($R^2$ increased from 0.0% to 51.7%, corresponding to a partial $R^2$ of 51.7%, F(22,1) = 23.6, $p$ = 0.00007). The results of this analysis are shown in **Fig 6A and 6B** where we illustrate the partial $R^2$ analysis by comparing each component of day 1 learning with the portion of 24-h retention not explained by the other (see Materials and methods for details). When we repeated this analysis using estimates of temporally-persistent and temporally-volatile adaptation based on either the first or the second measurement session alone rather than the averaged data, we found similar results (Session 1 only: partial $R^2$ of 43.1%, $p$ = 0.0005 for temporally-persistent adaptation versus partial $R^2$ of 1.0%, $p$ = 0.64 for temporally-volatile adaptation; Session 2 only: partial $R^2$ of 53.0%, $p$ = 0.00005 versus partial $R^2$ of 8.6%, $p$ = 0.16, correspondingly). This indicates that the measurements of temporally-persistent learning from both day 1 sessions independently predict subsequent 24-h retention on day 2, albeit with a nominally stronger association for the second session which was adjacent to the 24-h retention measurement.

In summary, we find in Experiment 4, that increased temporally-persistent adaptation is associated with stronger long-term memory, whereas increased temporally-volatile adaptation does not. This sharply contrasts with Experiments 1 and 2 where we found that temporally-volatile adaptation was associated with savings whereas temporally-persistent adaptation was not. Taken together, these results demonstrate a striking double dissociation between the contributions of temporally-persistent and temporally-volatile learning to long-term memory and savings. To even more directly compare the 2 contributions to this double dissociation, we returned to the savings data from Experiments 1 and 2 and performed a bivariate analysis of the inter-individual differences in overall savings based on the levels of temporally-volatile and temporally-persistent learning during retraining (see Materials and methods for details). This bivariate regression analysis is analogous to that performed on the 24-h retention data above and is illustrated in **Fig 6C and 6D**, in parallel format to **Fig 6A and 6B**. Adding temporally-volatile learning as a second regressor after temporally-persistent learning resulted in a

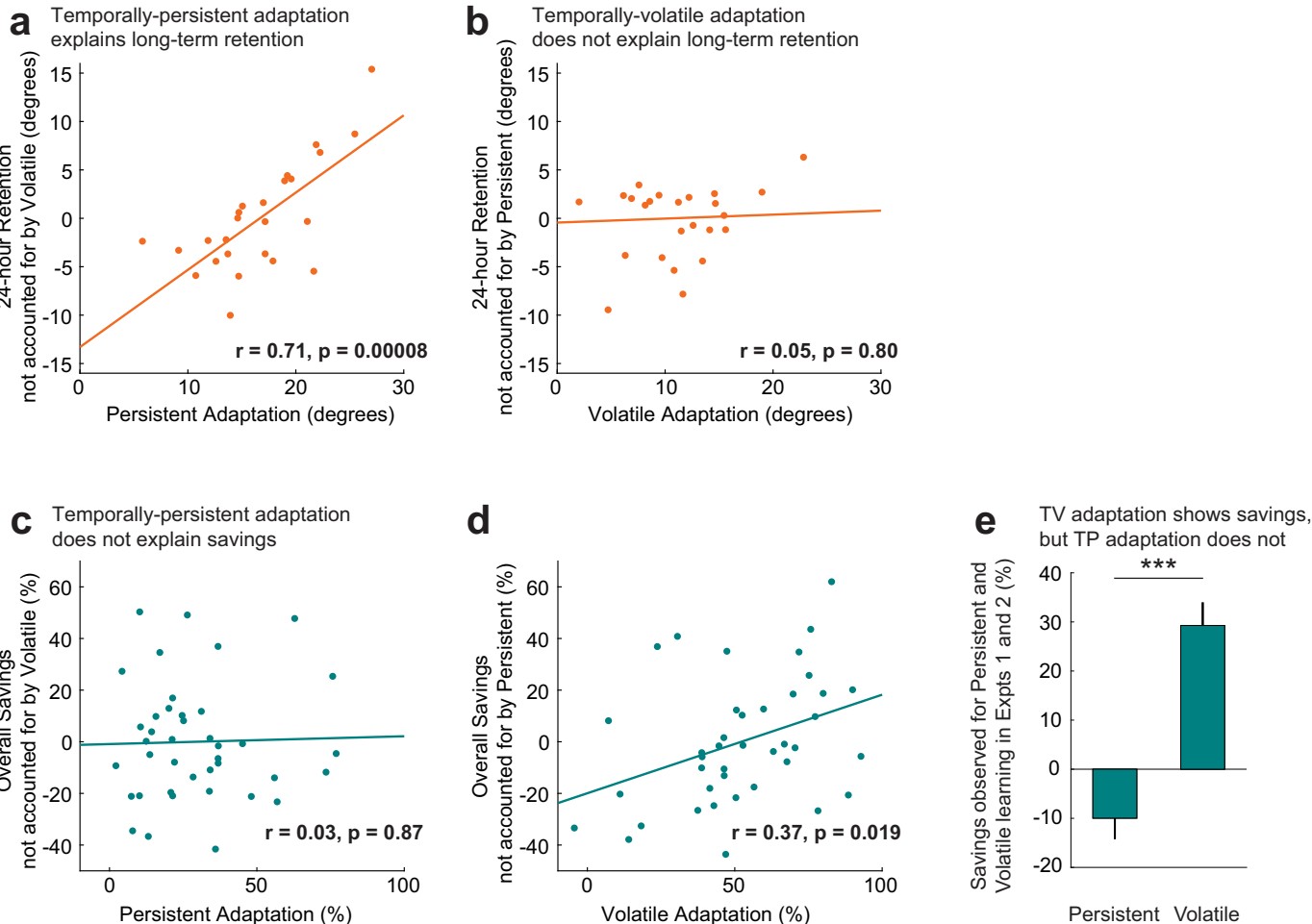

**Fig 6. A double dissociation between savings and long-term memory, uncovered by dissection of learning into temporally-persistent and temporally-volatile components.** **(a)** Illustration of the partial regression between temporally-persistent adaptation on day 1 (shown on x-axis) and 24-h retention across individuals ($N = 25$). The y-axis represents residuals of the univariate regression of 24-h retention upon temporally-volatile adaptation. The positive relationship indicates that higher temporally-persistent adaptation is associated with higher long-term memory. The solid line indicates linear fit. **(b)** Same as **a** but for temporally-volatile adaptation, showing no significant relationship (slopes restricted to positive values). **(c)** Illustration of the partial regression between temporally-persistent adaptation during relearning (shown in x-axis) and savings for early (trials 2–6) training combined across Experiments 1 and 2. The y-axis represents residuals of the univariate regression of savings upon temporally-volatile adaptation. There is no significant relationship. **(d)** Same as **c** but for temporally-volatile adaptation, showing a significant positive relationship and thus indicating that temporally-volatile adaptation explains overall savings. Note that mean y-axis values in **(a)**–**(d)** are zero, since they represent residuals of linear regression; these values do not reflect the actual amounts of long-term retention or savings which are, on average, significantly above zero. **(e)** Comparison of average savings for temporally-persistent (TP) and temporally-volatile (TV) adaptation, combined across Experiments 1 and 2. Error bars indicate SEM. ***$p < 0.001$. Underlying data supporting this figure can be found in files Exp_1_2_data.mat and Exp_4_data.mat.

significant improvement in the ability to explain savings ($R^2$ increased from 0.0% to 13.8% corresponding to a partial $R^2$ of 13.8%, $F(37,1) = 5.9$, $p = 0.0198$). We note here that this $p$-value, corresponds to the variance reduction associated with adding temporally-volatile learning to the regression and is equivalent to a two-tailed test for its regression slope being non-zero. Thus, if a one-tailed test would have instead been used, in line with the idea of testing for a significantly positive relationship between savings and initial temporally-volatile learning, its $p$-value would have been <0.01. In contrast, adding temporally-persistent learning as a second regressor after temporally-volatile learning resulted in no significant improvement in the ability to explain savings ($R^2$ increased from 13.7% to 13.8%, corresponding to a partial $R^2$ of

0.2%, F(37,1) = 0.1, $p$ = 0.81). The findings from this regression analysis show that interindividual differences in the temporally-volatile but not the temporally-persistent component of initial learning are associated with individual differences in the amount of savings during relearning. This adds to the evidence illustrated in **Fig 3**, summarized in **Fig 6E,** that temporally-volatile learning displays savings whereas temporally-persistent learning does not.

## Discussion

Here, we compared the mechanisms responsible for savings and long-term memory in human motor learning, finding that temporally-volatile adaptation leads to savings and that temporally-persistent adaptation leads to long-term memory. When we dissected adaptive responses into temporally-persistent and temporally-volatile components using 60-s delays (**Fig 1**), we found that temporally-persistent memories washed out 4 to 20× more slowly than temporally-volatile memories (**Fig 2**), leaving a considerable temporally-persistent residual even after 100 washout trials (see **Fig 2**), when overall adaptation had long since washed out to zero. This suggests that the short washout periods of 11 to 100 trials per trained movement direction used in a number of previous studies to wash out overall adaptation [13,15,16,18,24, 64] would likely have failed to wash out this temporally-persistent adaptation.

We, therefore, controlled for the effect of residual temporally-persistent adaptation, either experimentally with an extended 800-trial washout period which could eliminate it, or analytically with appropriate baseline subtraction and normalization, allowing us to accurately assess savings. With this control, we consistently found significantly greater savings for temporally-volatile than temporally-persistent adaptation (**Figs 3 and 6**). In fact, whereas temporally-volatile adaptation showed savings by displaying relearning that was consistently faster than initial learning, temporally-persistent adaptation remarkably showed anti-savings—displaying relearning that was significantly slower than initial learning both in the overall data and specifically in the 800-trial washout condition in which complete washout of both persistent and volatile learning occurred, allowing savings to be most cleanly measured. Temporally-persistent learning was also nominally slower, though not significantly so, in the 40-trial washout data. Remarkably, we found that savings in temporally-volatile adaptation was sufficiently large to overcome the anti-savings in persistent adaptation, and still confer robust savings on overall adaptation. Moreover, we found that the temporally-volatile savings we observed were due to implicit rather that explicit learning (**Fig 4**). Our data thus suggest that savings in overall adaptation is derived from implicit, temporally-volatile adaptation, representing an increased propensity to more rapidly form a temporally-volatile memory, rather than the reemergence of a temporally-persistent memory.

When we dissected adaptation into volatile and persistent components to examine the mechanisms for long-term memory, we found a strong positive relationship between 24-h retention and temporally-persistent but not temporally-volatile adaptation (**Figs 5 and 6**). Together, our findings for savings and 24-h retention delineate a powerful double dissociation whereby temporally-persistent learning leads to long-term memory but not savings and temporally-volatile learning leads to savings but not long-term memory.

### The juxtaposition between savings in temporally-volatile and temporally-persistent adaptation explains differences in savings across previous studies

Our findings provide a resolution to the apparent discrepancy between previous studies which isolated implicit savings in adaptation, yet found either anti-savings [45] or savings [20]. The paradigm used in the former study likely promoted temporally-persistent implicit adaptation,

which here we find to display anti-savings, because compared to the current findings which employed only 1 movement direction, the multiplicity of movement directions used would dramatically increase the temporal spacing between same-direction movements, limiting temporally-volatile adaptation—though the extent of this effect is unfortunately difficult to definitively evaluate because the inter-trial time intervals were not explicitly reported in the study. In contrast, the quickly paced paradigm used in the latter study would permit greater accumulation of temporally-volatile adaptation, which here we find to display enough savings to overcome anti-savings in temporally-persistent adaptation. Other factors may also have affected the balance between temporally-volatile and temporally-persistent components to drive the savings versus anti-savings observed in these 2 studies.

This effect whereby temporally-volatile savings would be reduced when a larger number of target directions are present in the experiment design, because same-direction inter-trial time intervals would increase and force temporally-volatile adaptation to decay to a greater extent, would also predict reduced savings even in studies that did not isolate implicit adaptation. This prediction is indeed borne out in previous work, with studies using 4- to 8-target paradigms finding either less pronounced savings [13,19] or no savings at all [24], whereas studies using 1-target or 2-target paradigms [15,17,18] demonstrating considerable savings. Given the approximately 20 s time constant observed for the decay of temporally-volatile adaptation during VMR learning [56–58], the fairly rapid 3-s inter-trial interval in our study would allow near-complete (86%) carryover of temporally-volatile adaptation from one trial to the next, whereas the 8-fold increase in inter-trial intervals expected for an 8-target study with similar trial pacing would allow only 30% carryover. This would dramatically reduce the effect of temporally-volatile savings.

## Incomplete washout can contaminate the assessment of savings

Our experiments revealed that temporally-persistent adaptation requires a surprisingly long period to wash out—well above 100 trials and much longer than overall adaptation that combines volatile and persistent components and is effectively washed out in just 20 to 40 trials (**Fig 2**). This occurs because a negative temporally-volatile adaptation acts to mask the enduring temporally-persistent component after 20 to 40 washout trials. Therefore, if the washout of temporally-persistent adaptation is not specifically measured, it is easy to get the false impression that a short block of 20 to 40 trials is sufficient to washout all adaptation so that true savings—which refers to relearning after complete washout of all adaptation—can be cleanly measured. In fact, most savings studies we examined employed washout periods well below what would be necessary for complete washout of temporally-persistent adaptation [13,15,16,18,24,64]—a notable exception being a recent [45] study which employed a long washout period with several breaks that probed for residual temporally-persistent adaptation. Incomplete washout of temporally-persistent adaptation would lead to an apparent savings due to the unmasking of this persistent memory during relearning as predicted by a multi-rate learning model [10], contaminating the assessment of true savings. Even Zarahn and colleagues [15], where the primary claim was that savings was present after the complete washout of all components of adaptation so that the savings they found could not be explained by apparent savings due to the unmasking of a slowly decaying learning process, used only 40 trials, meaning that the savings they observed almost certainly included apparent savings from incomplete washout of temporally-persistent adaptation. In fact, inspection of the Zarahn and colleagues data reveals that savings at trial 10 (the time point at which we found maximal savings) was about 20% of the perturbation size. However, we find that the short 40-trial washout that they used resulted in a residual level for temporally-persistent adaptation that was about

15% of the perturbation size, suggesting that the most of the savings they observed was apparent savings. The current results from the 800-trial washout data where temporally-persistent adaptation was entirely eliminated, do, however, show clear evidence for the true savings that Zarahn and colleagues hypothesized. It is unfortunately impossible to know precisely how much of the savings that study reported was due to incomplete washout because of paradigmatic differences, in particular their use of a 45-degree rather than a 30-degree VMR perturbation, that may have increased or decreased the degree to which their washout was incomplete or the amount of true savings present.

## Implicit savings does not result from the recall of a consolidated motor memory

Our findings dissociating savings from long-term memory, and demonstrating savings to be driven by temporally-volatile, rather than temporally-persistent memories, upend the widespread view that the faster relearning that characterizes savings results from the recall of a previously consolidated, stable motor memory [8,9,14,16,17,19,43,84,85]. Whereas savings has been taken as a litmus test for the consolidation of motor memory 1, 2, or even 7 days after training [13,24], here we find that savings is primarily driven by the temporally-volatile component of adaptation, which decays over the course of 1 min or less. These temporally-volatile savings could not arise from a consolidated memory of previous adaptation, as any memory solid enough to survive the hour-long, 800-trial washout period in our experiments, would certainly be stable against the passage of time during the minute-long delays in our experiment used to define temporal lability. Our findings thus indicate that savings is not driven by a consolidated memory of the adapted motor output, in line with the double dissociation between savings and long-term memory that we demonstrate.

In contrast, the savings we observe is driven by an increased learning rate for adaptive but temporally-volatile changes in motor output from one trial to the next when experiencing the same perturbation again after washout. While the ability to establish this learning rate increase could be considered a type of consolidated memory as it is long-lasting and must rely on a plastic change driven by experience, this ability is very different from the usual conception of a consolidated motor memory in which the trained actions themselves are remembered. Thus, our findings are consistent with a model in which implicit savings is based on an ability to improve the adaptation of actions, rather than a memory for the actions themselves. It is important to point out, however, that our findings do not show that consolidated memories are not formed or eliminate the possibility that they make some contribution to savings. For example, it is possible that explicit savings is driven by consolidated explicit memory. Even in our implicit adaptation data for which temporally-persistent relearning displays anti-savings, we cannot rule out the possibility that a consolidated temporally-persistent memory subcomponent exists and contributes to savings but that that the effect is overwhelmed by a larger anti-savings effect from a different subcomponent. Instead, our findings demonstrate that consolidated memories are not required for savings.

## Mechanisms for learning rate modulation in temporally-volatile adaptation

What are the mechanisms behind learning rate increases in temporally-volatile adaptation? Recent work suggests that such a learning rate increase can be driven by learning environments with increased statistical consistency, defined as a positive correlation between successive perturbations (i.e., lag-1 autocorrelation) in environmental dynamics or errors from one trial to the next [52–54]. This consistency-driven effect is further enhanced when the

environment repeats the same perturbation [54], with highly consistent, repetitious switching environments increasing learning rates up to 3× from baseline and highly inconsistent ones decreasing then up to 5×. Critically, the initial training periods in savings paradigms, including the current one, are characterized by both consistency and repetition, as they usually consist of a large number of trials with the same perturbation, leading to an increase in the learning rate during retraining.

## Savings and explicit adaptation

Experiment 3 revealed that the temporally-volatile savings we observe arises from implicit adaptation. This adds to recent evidence [19,20] against the idea that savings is exclusively driven by explicit adaptation [18,43–45]. Although savings can clearly occur because of explicit strategies, we did not observe this in our experiment. While this may, in part, reflect the considerable variability in the balance between implicit and explicit adaptation across individuals [74,86], the lack of explicit savings in our study was likely due to an experimental design that promoted implicit adaptation and minimized explicit strategy, and thus provided little power to detect explicit savings. The design elements included the lack of aiming instructions, the absence of workspace markers positioned to aid re-aiming, the use of point-to-point rather than shooting movements, and the engineering of low (approximately 25 ms) visual feedback latency for onscreen cursor motion, all of which may promote implicit learning [76,80–83]. In contrast, studies which reversed most of these design elements elicited primarily explicit adaptation and found clear explicit savings [44].

Taken together, the evidence now suggests that both implicit and explicit adaptation can show savings, albeit via different mechanisms. The current study shows that implicit savings is specifically driven by the faster relearning of a temporally-volatile memory, whereas previous work provides evidence that explicit savings is driven by temporally-persistent memory, as explicit savings is clearly observable in multi-target paradigms that would minimize temporally-volatile memory [44,45]. As a consequence, for experimental paradigms that primarily elicit explicit learning, we would expect savings to primarily be driven by this temporally-persistent adaptation, and for experimental paradigms that elicit a balance of implicit and explicit learning, we would expect the dichotomy between temporally-persistent and temporally-volatile contributions to savings to be blurred. However, because we did not examine these cases, we cannot know whether our expectations will be borne out or whether complex interactions between implicit and explicit learning might lead to different results that cannot be predicted from the current findings.

## Parallels between temporally-volatile/temporally-persistent learning and the fast/slow learning processes of motor adaptation

Another line of work has dissected motor adaptation, not experimentally, but instead on the basis of a computational model with 2 distinct adaptive processes: a fast adaptive process that learns rapidly and displays weak retention, and a slow adaptive process that learns slowly and displays strong retention [10]. By manipulating the training duration in order to elicit different amounts of the fast and slow learning, a subsequent study found that 24-h retention was specifically predicted by the amount of slow learning, rather by the amount of fast learning or overall adaptation [60]. Interestingly, this model-based dissection mirrors our temporal-stability–based dissection as the slow process, like temporally-persistent adaptation, leads to 24-h retention, whereas the fast process, like temporally-volatile adaptation, does not. In fact, there is a remarkable correspondence between the Joiner and colleagues study, which found that $49 \pm 6\%$ (95% confidence) of slow learning on day 1 is retained after 24 h, and Experiment 4 in

the current study, which found that 46 ± 9% of persistent learning on day 1 is retained after 24 h. Moreover, the trial-to-trial learning characteristics of the fast and slow processes mirror the ones for volatile and persistent adaptation, respectively. In particular, slow adaptation displays slower learning and better retention than fast adaptation, just as temporally-persistent adaptation displays slower learning and better retention than temporally-volatile adaptation (see **Figs 3C, 3D,** and **2A**, respectively). These parallels argue, although speculatively so, that the temporally-volatile and temporally-persistent learning from our dissection of adaptation corresponds to the implicit components of fast and slow processes from the two-state model. This possibility challenges 2 prominent ideas from the recent literature. First, the possible correspondence between the fast process and implicit temporally-volatile learning challenges the idea that fast process learning is synonymous with explicit adaptation [18,43,87]. Second, the possibility of a measureable instantiation of fast and slow process learning from the two-state model challenges the assertion that models of context-based learning and the switching between should supplant models of adaptive processes with different learning rates [41].

## The coexistence of temporally-volatile and temporally-persistent memories provides a mechanism for contextual interference

The possible mapping of temporally-volatile adaptation onto a fast learning/low retention process and of temporally-persistent adaptation onto a slow learning/high retention process provides an intriguing potential explanation for previous work on contextual interference in both motor and cognitive tasks. Contextual interference refers to phenomenon that memories formed in high-interference environments, where the task being performed is randomly switched from one trial to the next, are learned more slowly but show higher retention than memories formed in low-interference environments, where a single task is serially practiced [88–93]. If, as in the VMR adaptation task studied here, both temporally-volatile and temporally-persistent memories contribute to learning in tasks where contextual interference has been observed, then contextual interference effects can be predicted based solely on the temporal spacing inherent in the paradigms that elicit it. The idea here is that the high-interference condition in which tasks are randomly intermingled from one trial to the next would necessarily increase the temporal spacing between the trials within each task compared to the low-interference condition in which tasks are serially practiced. This increase in temporal spacing would allow temporally-volatile memories to decay, at least partially, and thus reduce the amount of temporally-volatile learning, which would slow the overall learning and promote increased temporally-persistent learning. Moreover, the resulting reduction in temporally-volatile learning and increase in temporally-persistent learning would both act to increase the proportion of learning that is temporally-persistent in the high-interference condition, which would, in turn, increase long-term retention according to the current findings. The slowed overall learning and increased retention predicted here for the high-interference condition are, in fact, the defining features of contextual interference. Correspondingly, the converse, faster overall learning but reduced retention, would be predicted for the low-interference condition where serially practiced tasks with reduced temporal spacing would allow temporally-volatile memories to rapidly build during training to improve performance, but would decay before a retention or transfer test, resulting in poor retention. Thus, the coexistence of temporally-volatile and temporally-persistent memories provides an explanation for contextual interference that does not require any interference itself. Further work will be required to determine the fraction of observed contextual interference effects that stem from this mechanism.

## Materials and methods

### Ethics statement

This study was approved by the Harvard University Committee on the Use of Human Subjects (CUHS). Participants were naïve with respect to the purpose of the experiments and provided written informed consent in accordance with CUHS policies.

### Participants

A total of 118 subjects (48 men, age 22.6 ± 4.7, 13 left-handed) participated in the present study (20 each in Experiments 1 and 2, 12 in Experiment S1, 41 in Experiment 3, and 25 in Experiment 4).

### Apparatus

We used the same experimental setup as the one used in recent work [62,63]. Subjects sat in front of an apparatus consisting of a 200 Hz digitizing tablet (Wacom Intuos 3 12" × 19", resolution of position data: 0.005 mm; accuracy: 0.25 mm) positioned below a 23" 120 Hz LCD monitor. During the experiment, subjects moved a custom-made handle, which contained a stylus, on top of the tablet allowing us to record hand position. Vision of the hand was occluded by the monitor and subjects instead observed their movement on the screen through a white cursor representing hand position.

### Experiment protocol

Using their dominant hand, subjects made point-to-point arm reaching movements between a starting position and targets 9 cm away. At the end of each movement, they were rewarded with a bell sound if they had managed to reach and stop at the target within 250 ms. Training was isolated to the outward movements, as visual feedback was unavailable during the return movements after the first 110 trials of the baseline block. Subjects took rest breaks roughly every 200 trials (about 7 to 10 min, see **Fig 1**).

Experiments 1 and 2 consisted of reaches towards a 90° target direction (in the midline, directly away from the body). After the 220-trial baseline block with no visual rotation, subjects in Experiment 1 ($N = 20$) entered the main part of the session which contained three 80-trial training periods. During training, a 30° VMR was imposed about the starting position. The sign of this VMR was the same for all training periods for each subject, with half the subjects training with a clockwise VMR and the other half training with a counter-clockwise VMR. The first and second training periods were separated by a 40-trial washout period, whereas the second and third training periods were separated by an 800-trial washout period. The training schedule in Experiment 2 ($N = 20$) was the same apart from that the 800-trial washout period came first (between the first and second training periods, see **Fig 1**).

Experiment S1 ($N = 12$) consisted of a longer, 800-trial baseline block with no visual rotation followed by a single 80-trial VMR training period like the ones used in Experiments 1 and 2.

Experiment 3 ($N = 41$) was similar to Experiment 2 in that it contained two 80-trial training periods separated by a 800-trial washout period (but not a third training period). It was designed to examine whether the temporally-volatile savings like the ones observed in Experiments 1 and 2 were due to an implicit or explicit process.

To dissect savings into implicit and explicit components, we used special instruction trials that prompted participants to disengage any explicit strategy by aiming their hand directly to the target. This method, also referred to as exclusion (since participants are to exclude strategies from their reach) [94], has been, in various forms, widely used to dissect implicit and

explicit visuomotor adaptation [44,73–78]. Specifically, instructions were given to either move to the center of the target or to its near/far end (both of which would not alter the reaching angle) and were presented immediately before and after the first (trial 10) 60-s time delay within both VMR training episodes (initial learning and relearning).

This enabled us to directly assess overall implicit adaptation (the amount of adaptation on the first instruction trial) and implicit-persistent adaptation (the amount of adaptation in the second instruction trial, which followed a 60-s delay), and, by comparing these 2, this enabled us to assess implicit-volatile adaptation. Both instruction trials had no visual feedback to avoid per-trial learning that would lead to posttrial recovery of adaptation. Moreover, by comparing adaptation in the second instruction trial to the no-instruction trial following it, we assessed explicit-persistent adaptation, and, by estimating overall adaptation as the average adaptation 2 trials before and after all these delay/instruction trials, we obtained estimates of overall explicit, volatile, and persistent adaptation (**Fig 4B**).

To minimize delays in reaction time, which would increase the inter-trial time interval and lead to further reduction in temporally-volatile adaptation, participants were presented with an "upcoming instruction" sound during the trial preceding the instruction. To familiarize participants with instruction trials (and the preceding "upcoming instruction" sound) ahead of VMR training, we presented a series of similar instruction trials during familiarization. Familiarization contained 4 different possible instructions: move your hand to the near, far, left, or right end of the (circular) target. There were clear biases towards the instructed end-points showing adherence to the instructions.

The aim of Experiment 4 ($N = 25$) was to examine the formation of long-term memories of VMR adaptation. The experiment began with a baseline period with no VMR that consisted of 456 trials, spread evenly across 19 target directions. After this baseline, subjects were trained on a 30˚ VMR for 120 reaches to a target placed at 90˚ (in the midline, directly away from the body, the same target used in Experiments 1 to 3). The direction of the 30˚ visual rotation was approximately balanced, with 13 subjects trained with a counter-clockwise VMR and 12 subjects with a clockwise VMR. This was followed by a testing block with 3 reaches towards each of the 19 targets, including the 1 target direction used throughout Experiments 1 to 3 (the other 18 directions were sampled to assess generalization of VMR learning as part of a separate study; here, we focus on learning and retention along the trained direction). During this block, visual feedback was withheld so that repeated measurements could be made without these measurements being contaminated by additional training that could be elicited by visual feedback. We used the movements towards the training direction to measure temporally-persistent adaptation. After this testing block, subjects were retrained on the 30˚ VMR for an additional 60 trials and after that were tested again without visual feedback to measure temporally-persistent adaptation as described above. Participants returned the following day to be tested for 24-h retention without visual feedback.

## Sample size determination

While sample sizes for experiment groups in analogous studies typically range between 8 and 12, here, we used somewhat larger sample sizes ($N = 20, 20, 41$, and 25 for Experiments 1, 2, 3, and 4, respectively; Experiment S1 had 12 participants, mirroring sample sizes in analogous studies). For Experiments 1 and 2, we examined a larger number of participants so that we could rigorously assess not only whether savings is present or not for temporally-persistent and temporally-volatile adaptation, but also the time course of savings for these 2 adaptation components at multiple points during training, as well as whether there are any subtle differences in savings or the extent of washout following the 40-trial versus the 800-trial washout

periods. The larger sample sizes in Experiments 1 and 2 also enabled more precise comparisons between the time course of washout for temporally-persistent and temporally-volatile adaptation, as the time constant estimates for these washout curves can be especially susceptible to noise in the data. In Experiment 3, we doubled the sample size relative to Experiment 2, given that Experiment 3 involved dissection of adaptation into 4 (explicit-persistent, explicit-volatile, implicit-persistent, and implicit-volatile), rather than 2 components. In Experiment 4, we examined $N = 25$ participants as we wanted to be able to look at not just the group-average amount of 24-h retention, but also examine how inter-individual differences in 24-h retention on day 2 related to inter-individual differences in temporally-persistent and temporally-volatile adaptation on day 1 (**Fig 4B and 4C**).

### Data analysis

**Statistical comparisons.** We performed single-sided paired $t$ tests across subjects to assess the presence of (positive) savings in adaptation and its subcomponents. For Experiment S1, which was designed to investigate potential mechanisms for anti-savings in temporally-persistent adaptation, we similarly used single-sided unpaired $t$ tests to compare Experiment S1 initial learning data to Experiment 1/2 800-trial washout relearning data. For all other statistical comparisons two-sided paired $t$ tests across subjects were implemented, with the exception of the comparisons involving the estimation of washout time constants in **Fig 2A** and the estimation of confidence intervals associated with the % contribution of temporally-persistent or temporally-volatile savings to overall savings: In these cases, we used a bootstrapping procedure (see below) instead of comparing fits to individual subject data, because the high noise in these individual data leads to low confidence about the corresponding individual parameters.

**Data inclusion criteria.** We performed outlier rejection on the learning curves of each experiment. Specifically, for each trial, we excluded adaptation levels that were more than 3 IQRs away from the subject median. This resulted in the inclusion of 99.4% of trials. Moreover, 1 participant in Experiment 3 was excluded from analysis due to inability to follow the experimenter's instructions.

**Estimation of visuomotor rotation adaptation.** To assess the amount of adaptation to the trained VMR, we measured the direction of hand motion on each trial. In movements with visual feedback, this was defined as the direction of the vector between the hand position at movement onset (based on a 6.4 cm/s velocity threshold) and the hand position 150 ms later. We used 150 ms to measure feedforward adaptation, as feedback corrections should be minimal at this point. In movements with no visual feedback used to estimate temporally-persistent adaptation and 24-h retention in Experiment 4, this was defined as the direction of the vector between the hand position at movement onset and the movement endpoint. To examine learning-related changes in performance, we subtracted out the small bias present in the baseline (0.13 ± 0.11˚) from all the movement-direction data.

**Measurement of temporally-persistent and temporally-volatile adaptation.** In Experiments 1, 2, 3, and S1, we measured temporally-persistent adaptation using 1-min delays interspersed with training. Because the temporally-volatile component of motor adaptation decays with a time constant of 15 to 25 s [56–58], the 1-min delays we impose here amount to 2.5 to 4τ, and thus lead to approximately 95% decay in temporally-volatile adaptation, effectively isolating the temporally-persistent component of adaptation. In contrast, the trial-to-trial decay in temporally-volatile adaptation for non-delay trials would be much lower, as the experiments were fast-paced with a median inter-trial time interval of 2.5 to 2.7 s, amounting to 0.1 to 0.2τ, thus leading to only 10% to 15% decay.

Thus, adaptation on the trial immediately following such a delay was operationally defined as temporally-persistent adaptation (**Fig 1D**). The corresponding overall adaptation as operationally defined as the average adaptation 2 trials before and 2 trials after the post-delay trial (with the exception of Experiment 3 which had additional trials to further dissociate adaptation into implicit and explicit components; see Experiment protocol section above). Temporally-volatile adaptation was taken as the difference between overall and temporally-persistent adaptation (**Fig 1D**).

These timed 1-min delays occurred every 30 trials during the VMR training blocks (on trials 10, 40, and 70 after the onset of each 80-trial training episode) and in 40-trial intervals during the long washout period, as shown in **Fig 1B and 1C**. During these delays, subjects held the handle still on the starting position. In addition to these timed 1-min delays, the Experiments contained rest breaks that allowed subjects to put the handle aside and were not strictly timed. These breaks occurred only during baseline or washout periods as shown in **Fig 1C**. We used the amount of adaptation after these breaks as a measure of temporally-persistent adaptation, but only when these breaks amounted to inter-trial intervals greater than 40 s (65.8% of these breaks for Experiments 1 to 3).

In Experiment 4, temporally-persistent adaptation was assessed during the no-feedback testing blocks that followed rest breaks (average break duration: 125 ± 8 s, minimum 58 s). Given a time constant for the decay of the temporally-volatile component of 15 to 25 s [56–58], this break would allow >99% decay in temporally-volatile adaptation and thus isolate temporally-persistent adaptation. Temporally-persistent adaptation was measured as the average of 6 reaches (3 reaches in each of the 2 no-feedback testing blocks) that were towards the training target. We designed Experiment 4 with 2 test blocks because we thought that averaging the data from both blocks might reduce the effects of measurement noise, as we expected that both temporally-persistent measurements would predict 24-h retention, but that the average might make a cleaner prediction. We estimated volatile adaptation as the difference between persistent adaptation and overall adaptation. The latter was assessed as the average adaptation during the last 20 trials of the training and retraining blocks. Finally, we calculated 24-h retention based on the no-feedback data from the testing block on day 2 (average of 6 reaches to the previously trained target, split over 2 consecutive blocks, **Fig 5A**).

**Estimation of washout time constants.** The washout of overall adaptation proceeded in 2 timescales: a very rapid initial washout phase during the first 2 to 3 washout trials, during which adaptation levels went from about 27° to about 11°, and then a slower washout phase that is illustrated in **Fig 1C**. To compare the time constants for washout for both temporally-persistent and overall adaptation (**Fig 2A**), we focused our overall washout analysis on the period beginning at trial 3 of washout in order to focus on the slower washout phase for comparing overall and temporally-persistent washout, because no temporally-persistent measurements were available during the very fast initial phase.

To estimate the values and confidence intervals associated with the time constants for washout, τ, we utilized a bootstrapping procedure [95]. Specifically, for each one of 10,000 bootstrap iterations, we randomly sampled, with replacement, $N = 20$ subjects from each group, and fit their average data with a single-exponential fit (**Eq 1**):

$$x = a + be^{-t/\tau} \tag{1}$$

When analyzing the overall washout curves, we discarded not only trials after each 1-min or rest break that removed the temporally-volatile component of adaptation in order to measure temporally-persistent learning, but also the 3 trials immediately thereafter, during which temporally-volatile adaptation might not be fully reequilibrated.

**Normalization of adaptation data.**   To systematically quantify savings, and specifically take into account the systematically different baselines between post-long washout versus post-short washout relearning, as well as the different baselines between temporally-persistent, temporally-volatile, and overall adaptation, we subtracted baseline adaptation, $x_{baseline}$, and normalized each learning curve $x$ by the distance between baseline and the ideal adaptation level of 30 degrees (Eq 2). The baseline level for overall adaptation was defined as the average of the last 5 trials before training onset, whereas the baseline level for persistent adaptation was defined as the average of the last 3 persistent-adaptation trials before training onset (in the case of baselines for initial training and training after an 800-trial washout) or as the last single persistent-adaptation trial, trial, 10 trials before the onset of training (in the case of baselines for training after a 40-trial washout, since the 40-trial washout contained only a single persistent-adaptation measurement trial).

$$x_{norm} = \frac{x - x_{baseline}}{30° - x_{baseline}} \times 100\% \tag{2}$$

**Estimation of savings.**   Finally, savings for each type of adaptation were taken to be the % difference in adaptation between initial training and retraining for the same training trial (Eq 3).

$$s = x_{norm}(retrain) - x_{norm}(initial\ training)\ (\%) \tag{3}$$

Throughout the study, we focused on savings around 1-min delay trials (especially trial 10 after training onset which captured early adaptation, but also trials 40 and 70), as these were the trials for which all 3 types of adaptation could be assessed. For the analysis of the across-individual relationships between savings and persistent/volatile adaptation (**Fig 6C and 6D**), however, because the measurement of temporally-volatile adaptation was based on the same measurements as overall adaptation (volatile = overall [adaptation 2 trials before and after the 1-min delay trial]–persistent [adaptation on the 1-min delay trial]), we instead calculated overall savings based on trials 2 to 6 (relative to rotation onset) in order to ensure that any observed relationships were not due to measurements shared between the dependent (savings) and independent (temporally-volatile adaptation) variables. This range was selected as it was both relatively far from the measurements used to calculate temporally-volatile adaptation, but also better captured the rapid rise of overall adaptation providing more power to assess inter-individual differences in savings.

**Comparisons of inter-individual differences.**   To examine contributions of temporally-persistent or temporally-volatile adaptation on savings and long-term memory (**Fig 6A–6D**), we used linear regression with slopes restricted to positive values to model positive contributions of these components of adaptation and either savings or long-term retention. Specifically, for studying long-term memory, we compared temporally-persistent and temporally-volatile adaptation on day 1 in Experiment 4 against 24-hour retention on day 2, whereas, for studying savings, we compared temporally-persistent and temporally-volatile adaptation from trial 10 in the retraining blocks in Experiments 1 and 2 against overall savings calculated as in the preceding paragraph.

## Supporting information

**S1 Fig. Anti-savings in temporally-persistent adaptation cannot be explained by prolonged reaching under baseline conditions.** Here, we investigated whether the anti-savings found in Experiments 1/2 (most pronounced following an 800-trial washout) might be due to the prolonged 800-trial washout period strengthening the baseline, unadapted state to the point that it

resists the formation of a temporally-persistent memory of adaptation during relearning. Previous literature suggests that this kind of repetition effect—a form of use-dependent learning [69,70]—tends to level off after only 50–150 trials [71]; thus, it should equally affect initial learning (which follows 220 baseline trials) and relearning after 800 washout trials, suggesting no net effect in the anti-savings we observe. However, this use-dependent learning effect has not been studied within the specific context of our task. Thus, in Experiment S1, we examined 12 new participants who adapted to a 30˚ visuomotor rotation following an 800-trial baseline period, to match the long washout period in Experiments 1 and 2. We found that the prolonged baseline in Experiment S1 (light gray) did not reduce the temporally-persistent component during adaptation compared to the shorter, 220-trial baseline in Experiments 1 and 2 (dark gray); instead, relearning after an 800-trial washout in Experiments 1 and 2 led to significant reductions in temporally-persistent adaptation, as we discuss in the main text. Together, these findings show that anti-savings in temporally-persistent adaptation were not due to the use-dependent learning during the long 800-trial washout period. **(a)** Comparison of average adaptation curves for (i) initial learning after 800 baseline trials from Experiment S1 (light gray), (ii) initial learning after 220 baseline trials from Experiments 1/2 (dark gray), and (iii) relearning after 800 washout trials from Experiments 1/2 (blue). **(b)** Close-up of the adaptation phase, with temporally-persistent measurements indicated by the empty circles as in Fig 3A. Note the similarity between the adaptation curves for the initial learning cases (after 220 and 800 trials of baseline) in contrast to the relearning curve. Error bars indicate SEM; red lines indicate 60-s delays used to isolate temporally-persistent adaptation. **(c)** Comparison of the levels of overall, temporally-persistent, and temporally-volatile adaptation for these 3 cases, at trials 10, 40, and 70 after the onset of the visuomotor rotation perturbation. Temporally-persistent adaptation displays no signs of reduction after the 800-trial baseline (new data) relative to the 220-trial one (Exp. 1/2 data); however, it is significantly higher than temporally-persistent adaptation during relearning after the 800-trial washout in Exp. 1/2. $^*p < 0.05$; $^{**}p < 0.01$. Underlying data supporting this figure can be found in files Exp_1_2_data.mat and Exp_S1_data.mat.
(PDF)

## Acknowledgments

We would like to thank Jasmine Bailey and Emerson Fang for their help with the experiments; Jordan Brayanov, Joel Greenwood, and Edward Soucy for their help with instrumentation; Rene Chen for the illustration in Fig 1A; Andrew Brennan and Yohsuke Miyamoto for helpful discussions.

## Author Contributions

**Conceptualization:** Alkis M. Hadjiosif, Maurice A. Smith.

**Data curation:** Alkis M. Hadjiosif.

**Formal analysis:** Alkis M. Hadjiosif.

**Funding acquisition:** Maurice A. Smith.

**Investigation:** Alkis M. Hadjiosif, J. Ryan Morehead, Maurice A. Smith.

**Methodology:** Alkis M. Hadjiosif, J. Ryan Morehead, Maurice A. Smith.

**Project administration:** Alkis M. Hadjiosif, J. Ryan Morehead.

**Resources:** Maurice A. Smith.

**Software:** Alkis M. Hadjiosif, Maurice A. Smith.

**Supervision:** Maurice A. Smith.

**Validation:** Alkis M. Hadjiosif, Maurice A. Smith.

**Visualization:** Alkis M. Hadjiosif, Maurice A. Smith.

**Writing – original draft:** Alkis M. Hadjiosif, Maurice A. Smith.

**Writing – review & editing:** Alkis M. Hadjiosif, J. Ryan Morehead, Maurice A. Smith.

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
