## [Editor Report · Decision Letter 0]

15 Aug 2022

Dear Dr Smith, 

Thank you for submitting your manuscript entitled "A Double Dissociation between Savings and Long-Term Memory in Motor Learning" for consideration as a Research Article by PLOS Biology.

Your manuscript has now been evaluated by the PLOS Biology editorial staff, as well as by an academic editor with relevant expertise, and I am writing to let you know that we would like to send your submission out for external peer review.

Once your full submission is complete, your paper will undergo a series of checks in preparation for peer review. After your manuscript has passed the checks it will be sent out for review. To provide the metadata for your submission, please Login to Editorial Manager (https://www.editorialmanager.com/pbiology) within two working days, i.e. by Aug 17 2022 11:59PM.

Kind regards,

Kris

Kris Dickson, Ph.D. (she/her)

Neurosciences Senior Editor/Section Manager

PLOS Biology

kdickson@plos.org

---

## [Decision Letter · Decision Letter 1]

2 Oct 2022

Dear Dr Smith,

Thank you for your patience while your manuscript "A Double Dissociation between Savings and Long-Term Memory in Motor Learning" was peer-reviewed at PLOS Biology. It has now been evaluated by the PLOS Biology editors, an Academic Editor with relevant expertise, and by several independent reviewers. 

In light of the reviews, which you will find at the end of this email, we would like to invite you to revise the work to thoroughly address the reviewers' reports.

Given the extent of revision needed, we cannot make a decision about publication until we have seen the revised manuscript and your response to the reviewers' comments. Your revised manuscript is likely to be sent for further evaluation by all or a subset of the reviewers.

**IMPORTANT - SUBMITTING YOUR REVISION**

*Re-submission Checklist*

*Published Peer Review*

*PLOS Data Policy*

*Blot and Gel Data Policy*

Sincerely,

Kris

Kris Dickson, Ph.D. (she/her)

Neurosciences Senior Editor/Section Manager

PLOS Biology

kdickson@plos.org

REVIEWS:

Reviewer #1: Speed of learning is faster for the tasks which has been learned before, known as savings. In motor learning which of the multiple motor learning systems is the main contributor to the savings is still under debate. In this manuscript, authors claim that the temporally volatile component of motor memory contribute to the savings, but the temporally persistent memory rather disrupts the savings. 

Participants learned a straight reaching under 30deg visuomotor rotation. After the learning, participants underwent a washout phase. A one-minute break was occasionally inserted during the washout phase, and the aftereffect measured immediately after this mini-break was defined as the component from the temporally persistent memory, meaning that it is robust to temporal perturbation.

Following are the main findings.

1) Time constant between temporally persistent and volatile motor memories indeed had different time constants. 

2) Even after 800 trials of washout, which eliminated the persistent component of memory (or at least brought the memory back to zero), the savings was still observed. Temporally persistent memory did not show savings.

3) Such savings existed when the task was designed to exclude any explicit component (i.e. "move your hand to the target" instruction), suggesting that the savings is not due to the explicit strategy.

4) Temporally persistent memory explains the long-term retention of the motor memory, but the temporally volatile memory did not. Conversely, temporally persistent memory did not explain the savings, but the volatile memory does. 

The authors conclude that the savings is not a manifesto of consolidated memory, but somehow arise from the increased learning rate when encountering the previously exposed perturbation pattern. The phenomenon itself is intriguing, and the result will point to a novel mechanism of how the motor memory is formed. The experiment is designed and carried out appropriately. However, I feel there are couple of points to be clarified before recommending for publication, especially about the interpretation of the data.

Major

1) The finding of this paper is that, there is an unknown mechanism which heightens the learning rate of the temporally volatile learning system during re-learning of the perturbation, but does not for (what they call) temporally persistent learning system. 

There must be some kind of long-term memory/storage mechanism which enables the savings to happen, but there is no guarantee that such long-term memory mechanism is captured by measuring the temporally persistent learning. Therefore, lack of signature of savings in the temporally persistent learning measure does not necessary means that the memory in the initial learning has not entered the long-term memory system and has not been consolidated. Thus, claiming that savings does not require consolidation of memory is a bit far stretched. 

To claim this, authors must show that the persistent memory measure indeed captures the sole consolidation mechanism for motor memory. Please clarify this point.

2) In Experiment 4, authors show that the temporally persistent memory indeed predicts the retention of the memory 24 hrs later. However, the way the persistent memory is characterized here is completely different to Exp 1-3. In Experiment 1-3, persistent memory was measured during the course of washout, thus represents the memory that is robust to the recent washout experience. However, in Experiment 4, it is not measured during the washout, thus, it does not represent the robustness against the washout experience. Also, the timing from the previous learning experience is not specified, and 19 other directions are tested together without visual feedback. Thus, experiment 4 is too different to the other experiments by design, so the data from this experiment is difficult to be used to validate the result of the Experiment 1-3 or even to be discussed together with them. Since result of this experiment is used as important evidence to claim for the double dissociation, I advise the authors to retest this with the same setup as in Experiment 1-3.

3) Authors showed that the temporally persistent component works in the opposite direction to savings during re-learning. However, the long washout period may have just simply made the persistent memory for the "baseline" and contributed to work in the direction of baseline, which is opposite to the direction of perturbation, during the re-learning. This is just like the persistent memory observed during the washout phase working in the direction of perturbation before being completely washed out. If this is the case, it will be difficult to say that the persistent component did not exhibit savings. Please discuss about this possibility. Perhaps the simple test would be to show how the persistent component is modulated during the initial learning, when preceded with baseline with different number of trials.

Minor

Line 286-293

Experiment 4 is typed as Experiment 3

Reviewer #2: The current behavioural study investigates across three visuo-motor adaptation experiments whether slow-and fast decaying adaptive responses to a previous rotation during wash-out is associated with different types of memory: 1) "savings" - the speed of re-adaption to a previously encountered visuo-motor rotation and 2) long-term retention - the presence of a consolidated motor adaptation memory 24h after acquisition. Slow and fast decaying adaption retention during washout and retraining were obtained following 60 sec break or trial-by-trial, respectively. Further explicit verbal instructions were used to manipulate implicit and explicit components of fast and slow decaying responses. The principal finding of this study is that fast decaying adaptive responses were associated with faster relearning (savings), but not with retention on the next day, challenging the assumed notion that savings are a marker of long-term memory formation. In contrast, slower decaying adaptive responses were related to long-term memory retention on the next day, but not with with relearning.

This study is interesting and timely - it questions whether the widely used measure of savings during relearning (obtained on the same day after wash-out) can be seen as a proxy for memory consolidation. My main concerns relate to the conceptual interpretation of the employed measures and a general lack of integration of the findings with literature outside the motor adaptation community (e.g. contextual interference).

1. The 60 sec break manipulation is very elegant and triggers persistent responses from a previous task condition, respectively, across washout and adaptation phases. Yet, it remains conceptually uncertain why it is framed as a component of motor learning and adaptation, rather than as a reflection of the prior movement history (use-dependent learning) and history of action selection (e.g. Valyear and Fitzpatrick 2018) here with respect to the movement direction. For instance, the response following 60 sec breaks during the first adaption period is biased in the reach direction during the baseline (no rotation). It is inverted in the direction of motor commands before any motor learning / adaptation has taken place, which challenges the idea that this is a measure of a newly formed motor memory (persistent or not). Accordingly, after the first visuo-motor rotation period the persistent response reflects bias in the opposite direction, due to the recent induced adaptation response history and then flips again in subsequent adaptations towards the washout response. The fact that the initial and relearning adaptation response after a 60 sec break show equal bias towards the preceding baseline/washed out response could be seen as simply reflecting recent movement and action selection history. Perhaps there is a way to tease out the conceptual difference by testing whether the persistent response is different following a newly formed adaptation memory, whether it is unique to this scenario. As a minimum the manuscript would benefit from clarifying whether and how these components can be disentangled conceptually in the context of this task.

2. The effect of volatile motor learning being associated with faster adaptation and temporally persistent motor learning (assessed following pauses) being associated with better memory retention on the next day echoes seminal older findings of contextual interference found across different motor (Battig 1972, Shea and Morgan 1979; Cross et al. 2007 and many others), as well as cognitive tasks (de Croock, van Merrienboer, & Paas, 1998). Here for blocked vs spaced-out practise of new skills similarly lead to faster acquisition vs long-term retention, respectively. The manuscript should outline how the current paradigm - in which volatile and persistent components of memory are obtained from the same session and within participants - relates and adds to this seminal literature.

3. There is no computational modelling employed in this manuscript, yet a direct link is postulated between volatile/persistent and fast/slow learning process of adaptation, respectively in the Discussion. Unless a computational model used by the authors in their previous work is employed, this claim remains speculative and this putative parallel should be tagged as such in the Discussion. 

4. In its current form, the manuscript is written to a specialised "niche" readership of motor adaptation researchers and full of jargon. I suggest to increase clarity by adding an explanatory figure, similar to the one in Fig. 4b to the beginning of the manuscript. For instance, it would be useful to outline the mapping between response measures surrounding 1-minute breaks, derived measures and how they map onto each memory component (e.g. volatile vs persistent) in the three experiments, respectively. This information is currently buried in the Methods section at the end of the manuscript. It should take a much more prominent position to increase understanding and critical evaluation by researchers outside the immediate specialist area from the outset of the manuscript. 

Minor comments:

* p. 5 line 126: missing p values

* p. 6 line 155 and 171: missing p values

Reviewer #3: In this study Hadjiosif and collaborators aimed at dissecting the learning mechanisms at the basis of savings and long-term memory in visuomotor adaptation. Through four experiments in which they manipulated washout duration, implicit/explicit learning and long-term memory, the authors find a dissociation between the mechanisms supporting savings and long-term memory. Contrary to what is generally believed in the field, they find that savings is explained by a temporally labile rather than a temporally persistent component of adaptation, suggesting that it may reflect meta-learning rather than memory strength. In contrast, they find that long-term memory depends on a temporally persistent component of adaptation. Interestingly, savings appears to arise from implicit, not explicit adaptation. 

The study is novel and certainly relevant to the field. It sheds light on several key aspects of motor learning using a simple experimental design via clean experimental manipulations and straightforward analyses. The reported dynamics on the decay of the temporally stable adaptation process, and its modulation by the amount of training are both critical findings. Until now, complete washout was estimated based on the overall level of performance. Tracking persistent adaptation using clever 1 min intervals interspersed through the washout period unveiled the true dynamics of memory decay, which are much lower than previously thought. Another important contribution of the study is the finding that savings is driven by the temporally labile component of learning. This is both interesting and, again, unexpected since this phenomenon has been naturally associated with the amount of training and the strength of memory both in the procedural and declarative fields. Finally, and in contrast to previous work, it shows that savings arises from the implicit learning process. Overall, the methodology and statistical analyses are sound, and easy to follow and the figures are clear, although some captions need improvement. Yet, at times the manuscript is hard to follow. Below, I list some suggestions and raise a few concerns that, if addressed, will help improve the quality of the work. 

1. The abstract is somewhat disorganized. The aim and hypothesis are there, but some key information is missing, for example the definition of the temporally volatile and persistent components of learning, which are key processes quantified here. It is stated that adaptation was experimentally dissected based on short-term (1-minute) temporal persistence. Yet, this sentence is not sufficient to understand what is being measured in terms of the overall aim of the study. Also, note that some sentences, like the results sentence, are long and intricate. If possible, reword for better comprehension. 

2. Overall, the Introduction is well written but does not emphasize the main aim enough, i.e. dissecting the learning mechanisms at the basis of savings and long-term memory. In fact, the literature background is focused on savings solely (there is in fact too much detail, some of which may be more appropriate in the Discussion). Given that the double dissociation between savings and long-term memory is a major contribution of the study, both the literature on savings and long-term memory should be considered in the Intro. Moreover, given the central relevance of the temporally volatile and persistent learning processes to the aim of the study, they should be defined in this section, and linked to the slow and fast modules. In the current version of the manuscript these processes are defined in terms of the experimental manipulation (persist vs doesn't persist the 1 min interval) not in terms of their function. This conceptual information is important to put the aim of the study in context.

3. The description of how volatile adaptation was quantified (lines 598-602 of the Methods) is cumbersome. Please provide a clearer explanation since it is key to understand the outcomes of the study.

4. Figure 1C shows that performance during adaptation doesn't seem to quite reach an asymptote in Experiment 1, which seems to be the case in Experiment 2. This may impact on the subsequent estimation of savings since there may still be room for improvement. How is this taken into account in the analyses? Data normalization won't help. Please comment on this potential limitation, and provide a quantification of the achieved asymptotic performance for these experiments. 

5. I understand that Experiment 4 aims at examining whether the temporally persistent and labile components of learning relate to long-term memory. Yet, what is the rationale for including two sessions of practice and having two measurements of the temporally persistent process? How is that an advantage when trying to link this process to retention, considering there would be cumulative effects of further training? Which measure of persistent adaptation was correlated to memory retention, the one corresponding to the first session, the second session, or their average? Were any of them more strongly correlated to long-term memory? I can't find this information in the Methods. 

6. I don't see how Figure 5 provides evidence supporting that Temporally-persistent adaptation leads to long-term memory, which is claimed in the figure's title. This seems rather more related to Figure 6, so please modify the title. Also, given that the results depicted in Figure 6 are not causal but correlational it would be more appropriate to refer to it as an association instead of assuming causality. The latter applies for the rest of the manuscript. In addition, the caption of figure 5 is somewhat confusing. What is being measured and represented here as temporally stable measurements? Is it the temporally persistent adaptation component? And how is it measured compared to Figure 1? Although stated somewhere in the Results section, neither the Methodology nor the figure caption indicate that temporally persistent adaptation was quantified 1-minute post training. Please provide all the necessary information in the figure caption, and use the same terminology to refer to the variables of interest consistently across figures, e.g., persistent adaptation instead of temporally stable measurement, etc. 

7. The conclusions drawn from this study may depend on the particular experimental paradigm/design chosen here. The authors claim that their task elicited more implicit than explicit learning. Would the results hold if the instructions were changed so that the task elicited relatively more explicit or equal amounts of explicit and implicit learning? In other terms, how generalizable are these findings to other paradigms/designs of VMA? This is a relevant issue and should be addressed in the Discussion. 

Minor

8. In line 212 there is a disconnected word: "Individually".

---

## [Decision Letter · Decision Letter 2]

16 Feb 2023

Dear Dr Smith,

Thank you for your patience while we considered your revised manuscript "A Double Dissociation between Savings and Long-Term Memory in Motor Learning" for publication as a Research Article at PLOS Biology. This revised version of your manuscript has been evaluated by the PLOS Biology editors, the Academic Editor and the original reviewers.

Based on the reviews and discussion with our Academic Editor, we are likely to accept this manuscript for publication, provided you satisfactorily address the the data and other policy-related requests at the bottom of this email, along with the few minor comments from Reviewer 1. Please note that failure to fully address all of these points will delay processing of your manuscript at resubmission. When resubmitting your manuscript, we'd also suggest you consider a title change to make this work more accessible for our broad biology audience. We'd suggest something along the lines of: Memory reacquisition savings reflects implicit adaptation rather than explicit reactivation of long-term memory (NB: I think "savings" is singular here so goes with "reflects"?)

We expect to receive your revised manuscript within two weeks. 

*Published Peer Review History*

*Press*

Sincerely,

Kris

Kris Dickson, Ph.D., (she/her)

Neurosciences Senior Editor/Section Manager,

kdickson@plos.org,

PLOS Biology

BLURB:

Please provide a blurb which, if the paper is accepted, will be included in our weekly and monthly Electronic Table of Contents (eTOCs), sent out to readers of PLOS Biology. This blurb may also be used to promote your article on social media. The blurb should be about 30-40 words long and is subject to editorial changes. It should, without exaggeration, entice people to read your manuscript, should not be redundant with the title and should not contain acronyms or abbreviations. For examples, view our author guidelines: https://journals.plos.org/plosbiology/s/revising-your-manuscript#loc-blurb

DATA POLICY:

Note that we do not require all raw data. Rather, we ask that all individual quantitative observations that underlie the data summarized in the figures and results of your paper be made available as this is essential to allow our readers to reproduce your data.

Thank you for providing access to your underlying data on GitHub. Unfortunately, we cannot accept sole deposition of data to GitHub or a similar non-static site (e.g. no personal sites and generally no institutional sites - see:(https://journals.plos.org/plosbiology/s/data-availability). We require deposition to a static site, like Zenodo, FigShare, OSF or provision as supplementary tables (i.e. excel). GitHub and similar sites can be used for depositing code however.

Please note that the data available on GitHub site can be easily copied to Zenodo. Once you do this, it will also generate a DOI number that you can provide us with. See the process for doing this here: https://docs.github.com/en/repositories/archiving-a-github-repository/referencing-and-citing-content

1) Please ensure that your online deposition or supplementary tables include the summary data for the following main and supplemental figures:

Figure 2A,B; Fig3A-F; Fig4A-C; Fig5A,B; Fig6A-E

Supplemental: Fig1A-C

2) Please ensure that figure legends in your manuscript include information on where the underlying data can be found (e.g. “The underlying data supporting Fig X, panel Y can be found in file Z.”).

3) Please also ensure that your supplemental data file/s has a legend.

4) Please ensure that your Data Statement in the submission system is updated to accurately describes where your data can be found.

DATA NOT SHOWN?

- Please note that per journal policy, we do not allow the mention of "data not shown", "personal communication", "manuscript in preparation" or other references to data that is not publicly available or contained within this manuscript. Please read over your submission carefully and either remove mention of any such data or provide figures presenting the results and the data underlying the figure(s).

Reviewer remarks:

Do you want your identity to be public for this peer review?

Reviewer #1: No

Reviewer #2: Yes: Katja Kornysheva

Reviewer #3: Yes: Valeria Della-Maggiore

Reviewer #1: The authors thoroughly and clearly answered all the concerns I have raised. Great respect to running the additional experiment. I have no further major comments. 

I went through the revised manuscript again, and I felt is indeed an interesting and enjoyable paper to read. It throws novel evidence to the debate on the mechanism of 'savings' observed in motor learning studies, proposes a different mechanism between savings and long-term memory, and also demonstrates the power of behavioural studies for capturing the novel processing of the brain. 

Minor: 

1) Line 152-153: I guess you are mentioning about the difference between Exp1 and Exp2, but is not clear in this sentences.

2) Perhaps I am still missing something, but why do you need reaching to 19 different directions in the baseline and testing blocks (lines 677, 681)? Is it done in other experiments also? If you were trying to see the generalisation but not presenting the data here, perhaps you should mention that in the method section, by saying that you focus on the primary purpose of the study; otherwise, will confuse the reader.

Reviewer #2: The authors have addressed all comments in a comprehensive and thoughtful revision, including an excellent point-by-point response. I have no further concerns or suggestions.

Reviewer #3: The authors have addressed all my concerns, and have implemented the suggested changes. I find the manuscript has improved significantly. It will make an important contribution to the field.

---

## [Editor Report · Decision Letter 3]

6 Mar 2023

Dear Dr Smith,

Thank you for the submission of your revised Research Article "A Double Dissociation between Savings and Long-Term Memory in Motor Learning" for publication in PLOS Biology. I have taken over its handling in the absence of Kris Dickson from the office, to avoid any unnecessary loss of time. On behalf of my colleagues and the Academic Editor, Raphael Kaplan, I am pleased to say that we can in principle accept your manuscript for publication, provided you address any remaining formatting and reporting issues. These will be detailed in an email you should receive within 2-3 business days from our colleagues in the journal operations team; no action is required from you until then. Please note that we will not be able to formally accept your manuscript and schedule it for publication until you have completed any requested changes.

I have seen your thoughtful response to Kris' suggestion of a title change and we are happy to keep your version.

PRESS

With best wishes,

Nonia

Nonia Pariente, PhD, 

Editor-in-Chief

PLOS Biology

npariente@plos.org